

# Fundamental effect of vibrational mode on vortex-induced vibration in a brimmed diffuser for a wind turbine

Taeyoung Kim[1], Hiroto Nagai[1], Nobuhide Uda[1], Yuji Ohya[2]

[1]Department of Aeronautics and Astronautics, Kyushu University, 744 Motooka, Nishi-ku, Fukuoka, 819-0395, Japan
[2]Research Institute for Applied Mechanics, Kyushu University, 6-1 Kasuga-koen, Kasuga, Fukuoka, 816-8580, Japan

*Correspondence to*: Taeyoung Kim (kim.ty07@gmail.com)

**Abstract.** A brimmed diffuser for a wind turbine, also known as a wind lens, is a ring-like short duct that is installed around a rotor. It gathers and accelerates wind to improve the power generation efficiency from the wind turbine, and this effect results from vortex shedding intentionally generated by the brim. However, periodic vortex shedding can induce a vibration 10  in the wind lens structure, which could potentially harm it in the case where resonance occurs when the vortex shedding frequency corresponds to the natural frequency of the wind lens structure. In this study, we investigated the fundamental mechanism of the vortex-induced vibration (VIV) in the brimmed diffuser structure at the Reynolds number of 288. A 2D aeroelastic analysis was conducted, utilizing 2D computational fluid dynamics coupled with the equation of motion in modal space based on the 3D FEM analysis. The 2D aeroelastic analysis provided a reasonable estimation of the critical wind 15  speeds for the actual VIV observed in the wind lens structure. Also, we clarified the vibrational modes critical to the VIV of the wind lens structure, which are the radial and rotational modes of the brimmed diffuser section. Both modes were accompanied by the circumferential bending oscillation of the support arms fixing the brimmed diffuser and were susceptible to the vortex shedding patterns.

## 1 Introduction

The improvement of power extraction from wind turbines is a critical issue in wind engineering. It is also an attempt to overcome the Betz limit, which defines the maximum efficiency of kinetic energy extraction from wind turbines as 16/27 (Betz, 1928). To enhance the power efficiency of horizontal axis wind turbines (HAWTs), the concept of a diffuser was devised (Foreman et al., 1977; de Vries, 1979; Igra, 1980), and wind turbines with the diffuser were termed diffuser-augmented wind turbines (DAWTs). The theory of the DAWTs is that it is possible to amplify the power output from wind 25  turbines by boosting the mass flow passing through the rotor with a diffuser, based on the principle that the power output is proportional to wind speed cubed. The diffuser for DAWTs is an annular wing whose cross-section is similar to a streamlined airfoil for an aircraft, and the rotor is placed inside the diffuser. When wind flows into the diffuser, the circulation around the airfoil-shaped cross-section of the diffuser is formed into a circumferential array along with the ring-like diffuser, causing the lift force to be generated inward. Consequently, more mass flow is induced, and wind speed



through the rotor increases. As a result, the augmented power output can be obtained in comparison with the conventional HAWTs, and the Betz limit is exceeded (Hansen et al., 2000; van Bussel, 2007).

A brimmed diffuser, also called a wind lens, was developed to effectively accelerate the wind speed at the rotor (Ohya et al., 2002; Ohya et al., 2004). A wind lens turbine (WLT) denotes the wind power generation system in which the wind turbine is installed inside the wind lens shroud. As shown in Fig. 1a, the wind lens features a large brim at the diffuser exit.

Unlike the diffuser for DAWTs, which has a cross-sectional shape similar to a streamlined airfoil, the cross-section of the wind lens is a cycloidal curve with a vertical straight line added at the end of the curve (illustrated in Fig. 1b). It is designed to intentionally disturb the airflow passing by the diffuser to generate vortex shedding behind the brim, causing the air pressure in the region behind the brim to decrease, and the wind passing through the rotor is drawn to the low-pressure region and is accelerated (Ohya et al., 2002). This effect not only contributes to enhancing the power output from the wind

turbine but also enables the wind turbines to operate at relatively low wind speeds (Abe et al., 2005; Ohya and Karasudani, 2010).

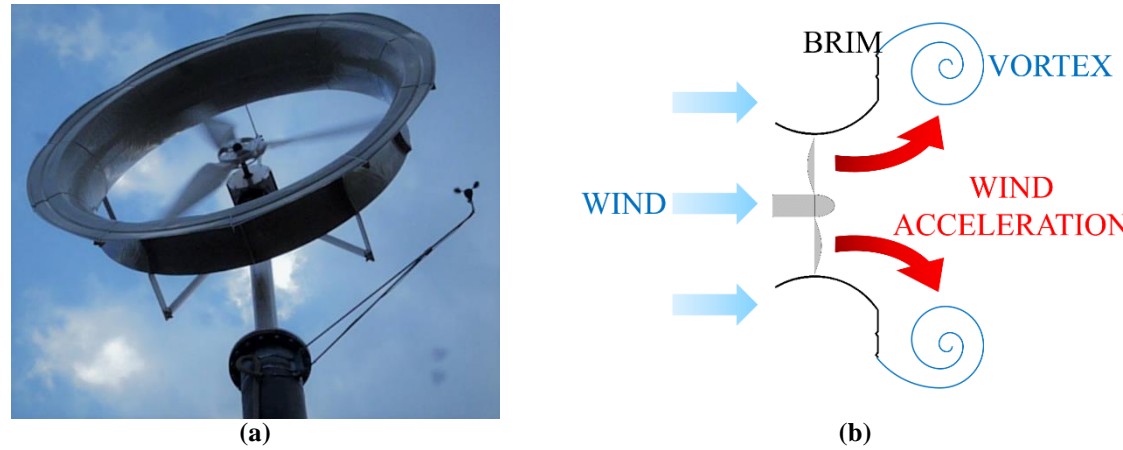

**Figure 1. A WLT filmed at Kasuga, Fukuoka, Japan on April 3, 2012 (a), and A schematic illustration of the principle of WLTs (b)**

On April 3, 2012, self-induced vibration in the brimmed diffuser for a 3 kW WLT installed at Kasuga, Fukuoka, Japan, was accidentally observed for approximately 15 minutes at 6:30 p.m. At the time of the observation, the weather was clear and windy, the average wind speed was 10.6 m/s, and a wind speed of 19.5 m/s was recorded as the maximum instantaneous wind speed (Japan Meteorological Agency, 2012). No damage was found in the wind lens after the noticeable vibration. However, this type of vibration caused by the wind could potentially harm the structure of the WLT system. Chou et al.

(2013) reported that vortex shedding from composite rotor blades for an HAWT caused a local resonance vibration in the wing edge, which resulted in the unexpected delamination and cracking of the composite blades during a typhoon. The maximum instantaneous wind speed of the typhoon was lower than the maximum wind speed of the wind turbine design. As this study suggests, the wind lens would not be an exception.



The interactions between fluid flow and structure have been observed in a variety of structures, including heat exchanger
tubes, tall buildings, and large bridges. The vibration phenomena induced by the fluid flow have interested many engineers
and researchers for decades, and fundamental studies on the flow-induced vibration (FIV) have been conducted for elastic or
elastically mounted circular cylinders (Sarpkaya, 1979; Bearman, 1984; Sarpkaya, 2004; Williamson and Govardhan, 2004;
Gabbai and Benaroya, 2005; Bearman, 2011), triangular prisms (Seyed-Aghazade et al., 2017), square prisms (Bearman,
1984; Bearman and Luo 1988), rectangular prisms (Parkinson, 1989; Matsumoto, 2008), semi-circular cylinder  (Zhao et al.,
2018), and H-beams (Schewe, 1989; Chen et al. 2012). These studies discuss the FIV for the corresponding bluff body. One
concern regarding the utilization of the vortex in WLTs is the structural vibration of the wind lens caused by the vortex
shedding. This phenomenon is known as vortex-induced vibration (VIV), which is an interaction between a structure in the
flow and a vortex shedding behind that structure. The common issue with the VIV, regardless of the cross-section of the
bluff body, is a *lock-in* phenomenon, which occurs once a vortex shedding frequency synchronizes to the natural frequency
of the structure. During the lock-in, the resonance vibration with a large amplitude occurs within a certain range of flow
velocity. For the elastic cylinders or prisms whose cross-section is non-circular, such as a square and a rectangle, there is a
possibility of experiencing a different type of vibration, termed galloping, whose motion is transverse or torsional with a low
frequency and large amplitude. This galloping phenomenon is velocity-dependent and has damping-controlled instability,
and is caused by the changes in the magnitude and direction of the unsteady flow-related force acting on the bluff body in
motion (Païdoussis et al., 2011). In the case of the single elastic circular cylinder, galloping does not occur because its
geometry with rotational symmetry does not produce the unstable motion-induced aerodynamic force (Zhao et al., 2018). A
characteristic of the galloping phenomenon is that its amplitude increases as the flow speed increases, while VIV exits the
resonance state when the vortex shedding frequency is desynchronized with the natural frequency of the structure. In this
respect, VIV and galloping are different; however, it is possible that these two types of FIV occur in combination (Bearman
and Luo 1988; Parkinson, 1989; Zhao et al., 2018). An experiment of FIV for a square prism conducted by Bearman and Luo
(1988) found that as the critical speed for galloping approaches the vortex resonance speed under a condition of low mass
damping, the amplitude of the oscillation triggered by the vortex lock-in increases with the increasing flow speed. Zhao et al.
(2018) experimentally investigated the FIV for an elastically mounted D-section cylinder whose section is semi-circular and
observed the VIV-galloping interaction in the case where the flat side of the D-section cylinder faces upstream. The flutter
phenomenon is also of concern. Experiments performed by Matsumoto et al. (2008) for a 2D rectangular prism, whose long
side is four times as long as the short side and is placed in parallel with the flow direction, found that the rectangular prism
experienced torsional flutter triggered by a motion-induced vortex, where the rotational axis of the prism is located at the
leading edge and there is minimal damping. The structure of the wind lens shown in Fig. 1a is basically different from that of
the elastically mounted cylinders or prisms mentioned above. The question posed is, "What kind of FIV did the wind lens
undergo at that moment?" It can be presumed that the self-induced vibration in the wind lens must have been primarily
involved in the vortex shedding since the geometry of the wind lens is intentionally designed to create large-scale vortices.



Motivated by the self-induced vibration observed in the wind lens, we initiated an investigation into FIV for the wind lens by numerical methods. The objective of this study is not only to simulate the realistic aerodynamic phenomenon of the actual wind lens but also to obtain the fundamental understanding of FIV for the wind lens. First, a 3D modal analysis of the wind lens structure was performed to determine the vibrational characteristics by using the finite element method (FEM). Next, representative natural modes that appear on the diffuser cross-section were extracted from the result of the 3D modal analysis. Then, a computational fluid dynamics (CFD) analysis for the flow around the wind lens structure and the fluid-structure interaction was performed. Considering the average wind speed on April 3, 2012, and the size of the wind lens, the WLT was in operation at a Reynolds number of approximately 300,000, which is in the critical Reynolds number regime. At a Reynolds number near 300,000 for a cylinder, the laminar boundary layer undergoes a turbulent transition, and the wake is narrowed and disorganized (Williamson, 1996; Anderson, 2007). Furthermore, the vortex shedding frequency is sensitive to the surface condition of the body and the turbulence intensity of the flow (Zdravkovich, 1990). Because of the complexity of the turbulence, the precise 3D CFD in the large Reynolds number regime is still challenging (Sarpkaya, 2004). Additionally, the difficulty of the direct numerical simulation (DNS) attributed to an extremely high resolution required to express the flow features, which demands an exceedingly high computational cost (Nguyen and Nguyen, 2016). Due to these factors, this study concentrated on the flow around the brimmed diffuser at a lower Reynolds number in two dimensions. This was done to reduce the influence of instability of the turbulent flow and the computational cost of the numerical simulation.

## 2 Three-dimensional structural model

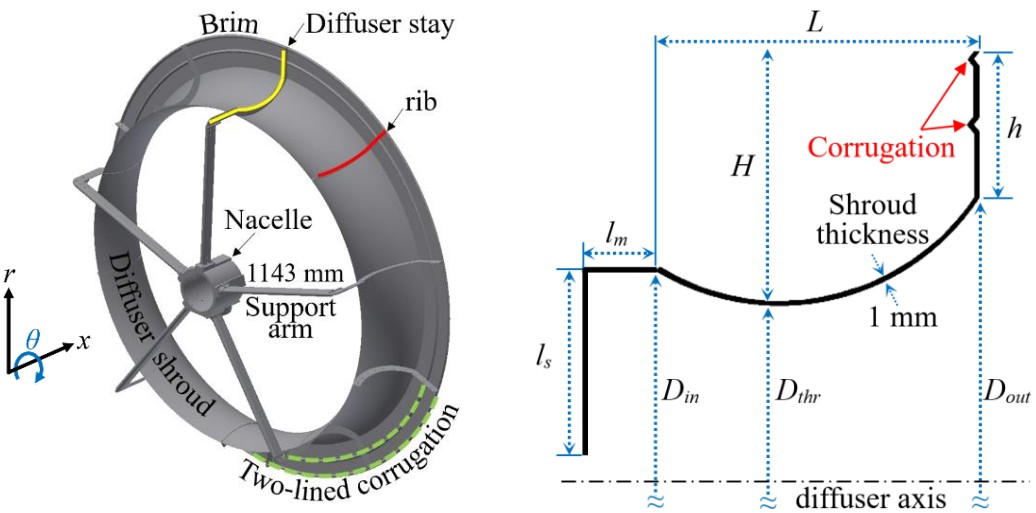

**Figure 2.** The 3D structural model of the wind lens (left) and a schematic illustration of the diffuser cross-section (right)





**Table 1. The dimension of each part of the brimmed diffuser in Fig. 2**

| Notation | Part | Dimension | Notation | Part | Dimension |
|----------|------|-----------|----------|------|-----------|
| $H$ | Diffuser height | 432 mm | $l_s$ | Support arm length | 1143 mm |
| $h$ | Brim height | 256 mm | $D_{in}$ | Inlet diameter | 2682 mm |
| $L$ | Diffuser length | 565 mm | $D_{thr}$ | Throat diameter | 2560 mm |
| $l_m$ | Straight member length | 300 mm | $D_{out}$ | Outlet diameter | 2912 mm |

The object of this study is the wind lens for 3 kW wind power generators, as explained in Sect. 1. The wind lens model
illustrated in Fig. 2 is named $C_{ii}B10$ where $C_{ii}$ represents one of the cycloidal cross-sectional shapes for a wind lens and B10
indicates that the height of the brim is 10% of the throat diameter of the wind lens (Ohya and Karasudani, 2010). The
diffuser length, $L$, is the horizontal distance from the inlet to the outlet, and the diffuser height, $H$, is the vertical distance
from the throat to the top of the brim tip. Both dimensions affect the flow around the wind lens; however, the diffuser height
is treated as the reference length of this model throughout this paper because the brim, whose dimension is included in the
diffuser height, is the key part of this *brimmed* diffuser. The above-mentioned dimensions of the parts are tabulated in Table
1. The material of the diffuser is Al6061 ($E$ = 68.9 GPa, $\rho$ = 2700 kg/m³, $v$ = 0.33). It is supported by five support arms
located 300 mm apart from the diffuser inlet, fixed to the nacelle. The support arms have a hollow rectangular cross-section
(the outer size: 89 mm in the $x$-direction×35mm in the $\theta$-direction, the inner size: 60 mm×25 mm) made of the same
aluminum alloy as the diffuser shroud. The support arms are installed with the long sides of the rectangular section placed in
parallel to the flow direction to prevent the wake generated by the arms from affecting the turbine blades. Five diffuser stays
firmly connect the diffuser shroud to the support arms. The diffuser stays are made of SS400 ($E$ = 206 GPa, $\rho$ = 7900 kg/m³,
$v$ = 0.3). They are 7.765 mm thick, and their width is 55 mm. Additionally, two types of structural reinforcements are applied
to the diffuser shroud. As shown in Fig. 2, ribs, whose material is also the steel, are mounted at the center between the two
neighboring diffuser stays to prevent deformation of the shape of the diffuser cross-section in the brimmed-diffuser shroud.
The shape of the ribs is the diffuser stay absent from the straight member to connect the support arm. The second
reinforcement is a two-lined corrugation added on the brim to prevent the brim from bending. The section of the corrugation
is an isosceles triangle with a 25.6 mm base and a 10 mm altitude.

## 3. Numerical methods

The FIV is an aeroelastic phenomenon, which is a coupling between structural and fluid dynamics. The aeroelastic analysis
consists of two steps. First, modal analysis is conducted to calculate the modal characteristics of the 3D wind lens structure,
such as natural frequencies and mode shapes. Second, the 2D numerical flow simulation is performed, coupled with the
modal equation of motion of the diffuser.



### 3.1 Modal analysis for the three-dimensional structure

The modal analysis was conducted for the 3D structure of the wind lens model, utilizing the commercial FEM software,
ANSYS 19.2. This analysis, based on the block Lanczos method (Rajakumar and Rogers, 1991), calculated the natural
frequencies and mode shapes of the wind lens model. The 3D FEM model, presented in Fig. 3, consists of four-node
rectangular shell elements for the diffuser shroud, diffuser stays, and ribs, and two-node beam elements for the five support
arms. The size of the elements is 7.5 mm, which was determined after the repetitive mesh dependency check. Thus, the total
number of elements is 192,565. The boundary condition is the bottom of the five support arms, which is fixed where the
support arms are fastened to the nacelle. The reference cross-section on the reference plane, shown in Fig. 3, was determined
to define a representative 2D mode shape of each mode for the 2D aeroelastic analysis. The reference plane is positioned at
the cross-section where the maximum displacement of each mode appears.

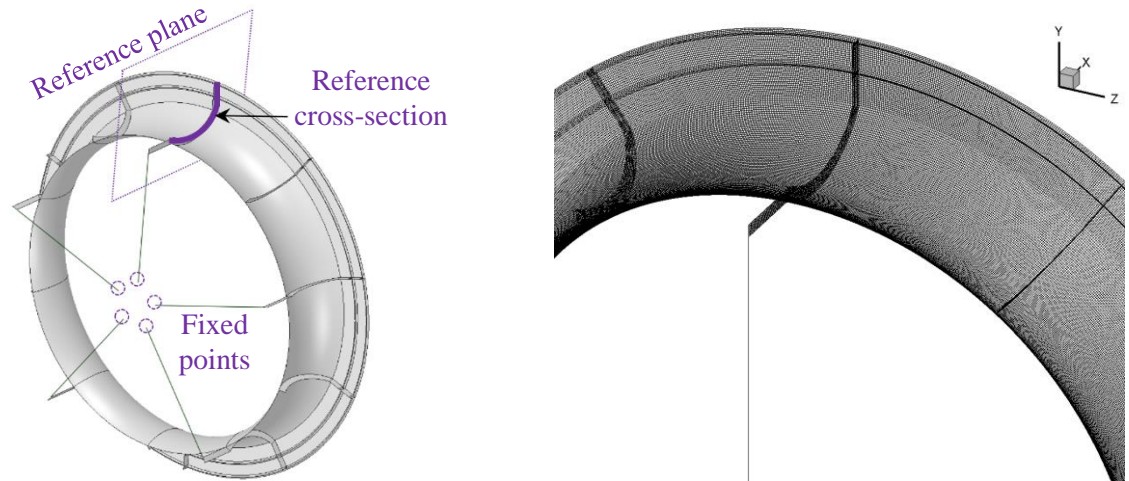

**Figure 3. The 3D structural model for modal analysis (left) and the elements in the model (right)**


### 3.2 Two-dimensional aeroelastic analysis

The 2D aeroelastic analysis was conducted utilizing the natural frequencies and the cross-sectional mode shapes on the
reference plane calculated in the 3D FEM analysis. The displacement vector $\{\boldsymbol{d}\}$ of the diffuser cross-section is represented
as a superposition of the four mode shapes as follows,

$$\{\boldsymbol{d}(\xi,t)\} = \sum_{k=1}^{4} \{\boldsymbol{\Phi}_k(\xi)\}q_k(t) \tag{1}$$





where $\{\boldsymbol{\Phi}_k\}$ is the $k$-th mode shape vector on the reference plane, $q_k$ is the generalized coordinate of the $k$-th mode, $\xi$ is the curvilinear coordinate along the diffuser cross-section. Lagrange's equation of motion in modal space for the $k$-th mode of the diffuser is expressed as follows,

$$\ddot{q}_k(t) + g_k\omega_k\dot{q}_k(t) + \omega_k^2 q_k(t) = \frac{Q_k}{M_k}, \qquad Q_k = \int_{L.E.}^{T.E.} \{\boldsymbol{\Phi}_k(\xi)\} \cdot \{\boldsymbol{\Delta P}(\xi,t)\}d\xi \qquad (2)$$

where $M_k$ is the generalized mass, $g_k$ is the structural damping coefficient ($g_k = 0.01$ in this study), $\omega_k$ is the natural circular frequency of the $k$-th mode, and $t$ is time. The generalized force, $Q_k$, is given by the integration along $\xi$ from the leading- and

trailing edge, where $\{\boldsymbol{\Delta P}\}$ is the aerodynamic pressure vector on the diffuser surface obtained with the 2D-CFD at each time step. Since the dimension in the mode shapes was reduced from 3D to 2D, the stiffness of each 2D mode was consequently slightly less than that of the actual 3D mode, which implies safe-side estimation. The equation of motion was numerically time-integrated with the implicit Houbolt method. To calculate the aerodynamic pressure distribution on the diffuser surface, an in-house CFD program including Eqs. (1) and (2) was utilized. The 2D compressible Navier-Strokes equation was

numerically solved with a finite difference method based on an implicit time-integration scheme (Steger, 1978) and a total diminishing variation (TVD) scheme (Yee and Harten, 1987). The detail of the computation is provided in the previously published studies (Isogai et al., 2004; Nagai et al., 2009). The 2D computational domain was $100L \times 80L$, as shown in Fig. 4a. The inflow boundary condition with the steady uniform flow of $U_0$ was applied to the left and top of the domain, and the right was in an outflow boundary condition. The bottom line, which coincides with the axis of the annular diffuser, was in a

symmetrical boundary condition, and the no-slip condition was applied on the diffuser surface. An H-type structured grid system was employed around the diffuser cross-section as shown in Fig. 4b, and these grids fitted and moved together with the deformed diffuser cross-section at each time step. The two corrugations on the brim, which reinforce the brim's bending stiffness, were neglected in the CFD model because they do not affect the flow around the diffuser. As explained in Sect. 1, the actual brimmed diffuser is 3D and is subject to the turbulent flow. To diminish the instability of turbulence, the Reynolds

number of the current brimmed diffuser model was reduced to 288 by modifying the dynamic viscosity in the 2D numerical simulation. Zhao et al. (2014) performed the 3D numerical simulations for a circular cylinder, whose span is 9.6 times the section diameter, and reported that the vortex shedding from the 3D cylinder is still 2D at a Reynolds number of 250. In other words, the vortex shedding pattern from any circular section of the cylinder is the same at a Reynolds number of 250 because there is no spanwise vortex shedding. Therefore, the 2D numerical simulation of the flow around the diffuser cross-

section performed for this study is valid at the low Reynolds number near 250. The similarity of the aeroelastic response is confirmed between the original and viscosity-scaled models due to the same mass ratio. A turbulence model was not employed in the CFD simulation at a Reynolds number equal to 288, which corresponds to a quasi-direct numerical simulation (QDNS) with fully fine grids and time step with a numerical dissipation from the schemes. The total number of the grids in the computational domain was 732,451 (1,439 in the $x$-direction and 509 in the $r$-direction), and 321 on the

diffuser cross-section. The grid system was divided into six blocks for parallel computation. The average mesh size inside





and behind the annular diffuser was $4.9 \times 10^{-3}H$ ($= 0.34\eta$) in the $x$-direction and $8.8 \times 10^{-3}H$ ($= 0.61\eta$) in the $r$-direction, and the mesh size adjacent to the diffuser surface was $1.6 \times 10^{-3}H$ ($= 0.11\eta$), where $\eta$ is the Kolmogorov length scale defined as $\eta \approx (v^3H/U_0^3)^{1/4}$. The time step was $7.8 \times 10^{-4}H/U_0$, which corresponds to the Courant number of 0.50, 0.013 of the Kolmogorov time scale defined as $(vH/U_0^3)^{1/2}$, and 1/6580 of the first vortex period defined as the first Strouhal number, which is

described subsequently. The calculation was conducted at each wind speed until the 80th cycle of the first vortex frequency. The aerodynamic force acting on the diffuser surface was sorted into the lift (in the negative $r$-direction) and drag (in the $x$-direction). The time-variant forces and displacement were evaluated from 27th to 80th cycles when a limit cycle oscillation appears. The employed grid system and time step were confirmed to be fully converged with respect to the frequency spectrum of the lift and drag acting on the diffuser. Details about the mesh and time-step dependency are discussed in

Appendix A.

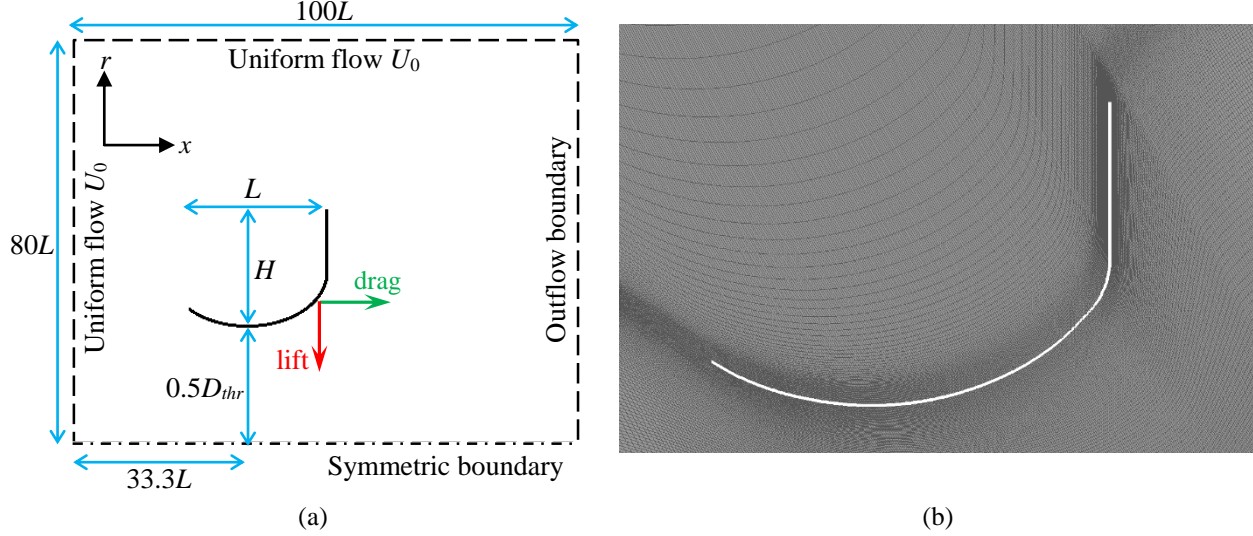

**Figure 4. The grid system of the 2D CFD: (a) The schematic of the computational domain, and (b) the grid near the wind lens drawn in white**

## 4 Results and Discussion

### 4.1 Modal characteristics

The 3D mode shapes of the wind lens model are complicated, making them difficult to define. For this reason, the 2D mode shapes, which are chosen from the reference cross-section where the maximum displacement occurs in each mode, are used to represent each 3D mode shape, as mentioned in Sect. 3.1. The interaction between the 2D modes and the flow around the reference cross-section triggers FIV. From the results of the modal analysis, we classified the 2D vibrational modes for the

wind lens structure into five basic patterns: radial, rotational, horizontal, brim-bending, and camber-bending modes, which are shown in Figs. 5–9, respectively.



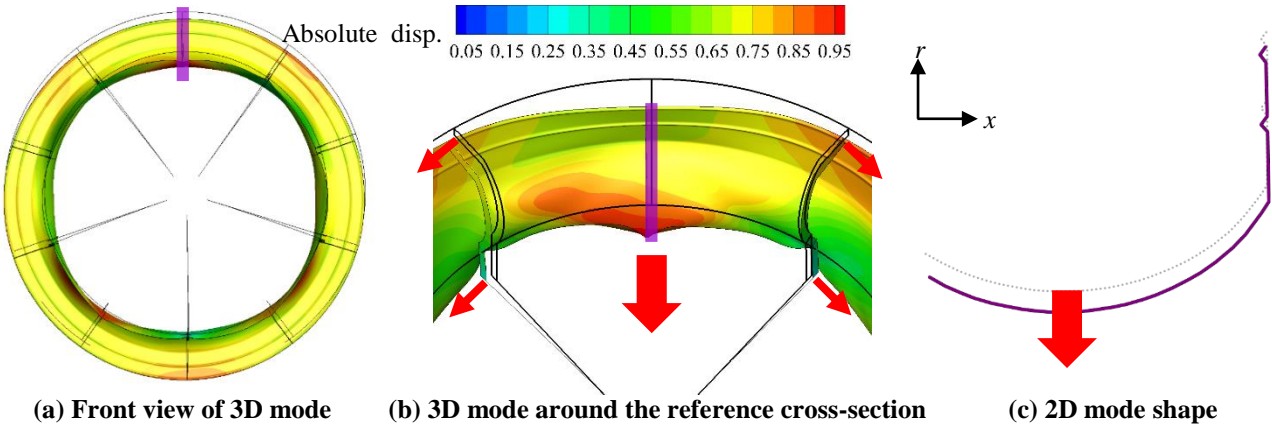

**(a) Front view of 3D mode**     **(b) 3D mode around the reference cross-section**     **(c) 2D mode shape**

**Figure 5. Radial mode for the 1st mode (degenerate mode)**

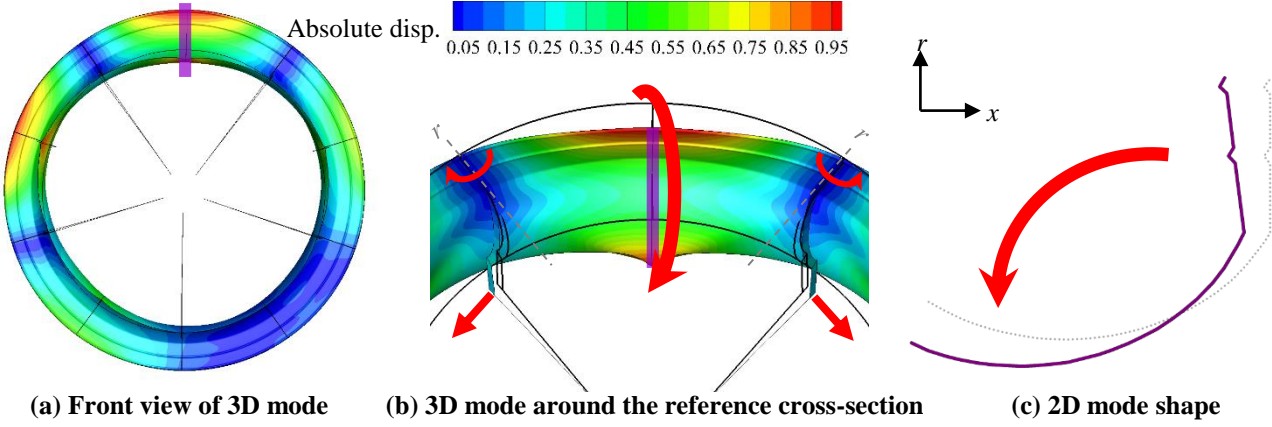

**(a) Front view of 3D mode**     **(b) 3D mode around the reference cross-section**     **(c) 2D mode shape**

**Figure 6. Rotational mode for the 2nd mode (degenerate mode)**

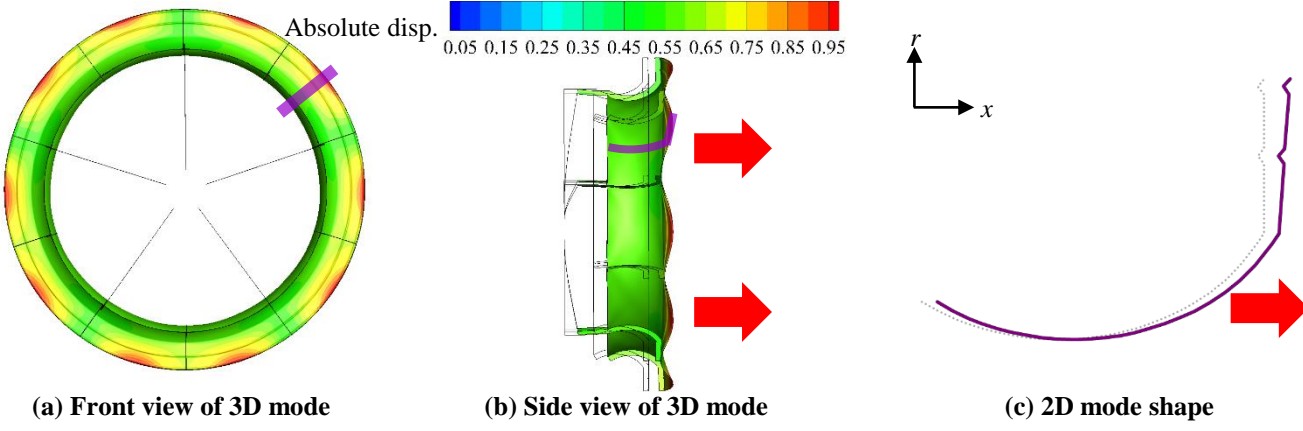

**(a) Front view of 3D mode**     **(b) Side view of 3D mode**     **(c) 2D mode shape**

**Figure 7. Horizontal mode for the 4th mode**



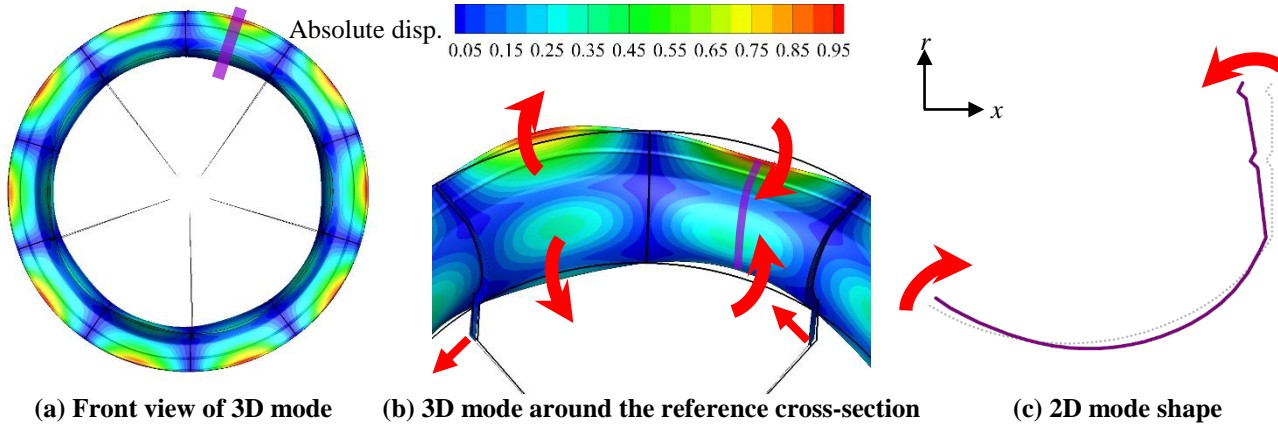

| (a) Front view of 3D mode | (b) 3D mode around the reference cross-section | (c) 2D mode shape |

**Figure 8.** Camber-bending mode for the 6th mode (degenerate mode)


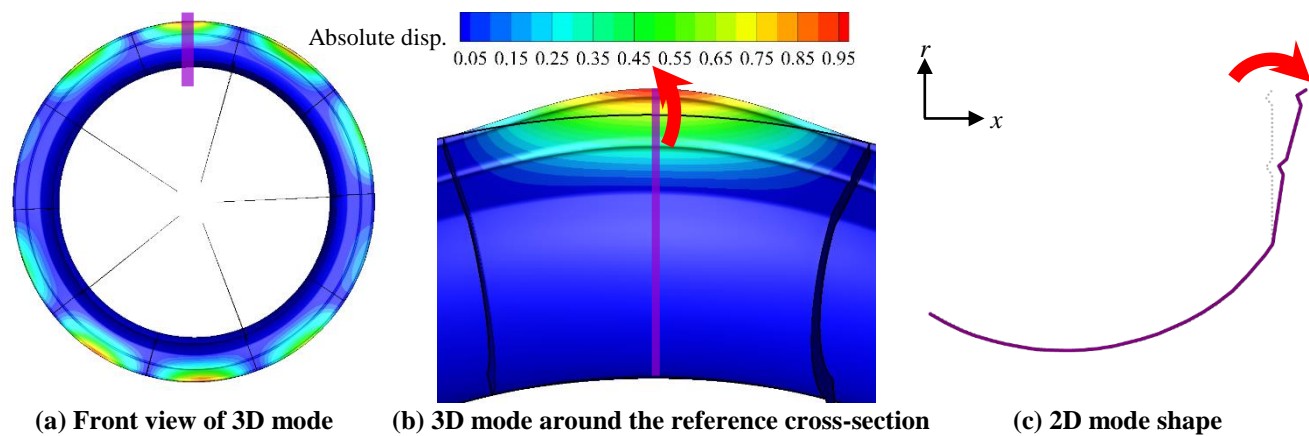

| (a) Front view of 3D mode | (b) 3D mode around the reference cross-section | (c) 2D mode shape |

**Figure 9.** Brim-bending mode for the 7th mode (degenerate mode)

The descriptions of the displacements are purposely exaggerated to illuminate the behavior of each mode. The contours
represent the absolute displacement, and the red arrows in the enlarged view indicate the direction of the motion. The
reference cross-section of each 3D mode is denoted by a short purple line on the image. The radial mode shown in Fig. 5 is
an oscillation of the reference cross-section in the radial direction, which is the typical transverse motion with respect to the
flow direction observed in the VIV of other sectional shapes, such as circular and rectangular. This mode shape is
accompanied by the out-of-phase oscillation of the neighboring support arms and stays in the circumferential direction.
When the interval between the neighboring support arms becomes wider due to their opposite circumferential bending, the
reference cross-section moves down and vice versa. The rotational mode, in which the reference cross-section rotates on the
reference plane, is demonstrated in Fig. 6. The neighboring stays pivot around the $r$-axis at the brim in the opposite direction,
and accordingly, the neighboring support arms undergo circumferential bending in the opposite direction. Therefore, as



shown in Fig. 6, the reference cross-section rotates as the interval between the leading-edges of the neighboring stays
repeatedly becomes wider or closer. The horizontal mode shown in Fig. 7 is accompanied by all five support arms bending in
the flow direction without circumferential bending. The camber-bending mode in Fig. 8 is the oscillation of the camber line
of the diffuser cross-section, which accompanies a minimal in-phase circumferential oscillation of the neighboring support
arms. The brim-bending mode in Fig. 9 is accompanied by the simple bending of the brim. Most of the 3D modes in the
wind lens structure have another degenerate mode, which has a similar mode shape but has the maximum displacement at a
different location. For this reason, in such a degenerate mode, vibration does not occur in the entire diffuser sections but a
specific part of the diffuser.

All possible mode shapes in the wind lens structure can be expressed with one or more modes out of the five basic mode
shapes. The natural frequencies and mode shapes from the first to fourth modes of the wind lens are tabulated in Table 2.

**Table 2. The mode shapes and the corresponding natural frequencies of the wind lens structure**

|  | 1st natural mode | 2nd natural mode | 3rd natural mode | 4th natural mode |
|---|---|---|---|---|
| **Mode shape** | Radial | Rotational | Camber-bending + Horizontal | Horizontal |
| **Natural frequency** | 10.92 Hz | 11.58 Hz | 20.85 Hz | 23.64 Hz |

The radial and rotational modes, which are accompanied by the circumferential bending of the support arms, easily occur
at the low-order natural frequencies, as shown in Table 2. The difference between the first and second natural frequencies is
small because both modes have a similarity in the out-of-phase oscillation of the neighboring support arms. It is noticeable
that the occurrence of the horizontal mode distinguishes the third and fourth natural frequencies from the first and second
ones. Namely, the horizontal mode accompanied by the bending of the support arms in the flow direction is less likely to
occur than the radial and rotational mode shapes accompanied by the circumferential bending of the support arms. As
described in Sect. 3.1, the long side of the cross-section of the support arm is installed in parallel to the flow direction so as
not to disturb the flow into the turbine blades. Accordingly, the bending stiffness of the support arms in the flow direction
can be raised by increasing the width in the $x$-direction; however, due to the limitation on the aerodynamic design and the
structural weight, it is difficult to increase the thickness in the circumferential direction. The third natural mode, in which the
camber-bending and horizontal modes are mixed, has a lower natural frequency than the fourth one because a small in-phase
circumferential oscillation of the neighboring support arms is accompanied by the camber bending. On the other hand, there
are no pure camber-bending (Fig. 8) and brim-bending modes (Fig. 9) in Table 2 because the ribs mounted on the diffuser
shroud and the two-lined corrugation on the brim have a preventative effect on such diffuser deformation modes. In
summary, the radial and rotational modes, which are related to the circumferential bending of the support arms, are easy to
occur and difficult to prevent because reinforcement of the support arms in the circumferential direction is constrained by the
aerodynamic and structural limitations.



## 4.2 Estimation of critical wind speed

The vortex shedding behind the brim is periodic and applies a time-varied load to the wind lens structure. When the vortex shedding frequency corresponds with the natural frequency, a resonant oscillation in the wind lens structure is likely to occur. The wind speed at which the resonance occurs is defined as the critical wind speed. In the case where the wind lens structure is at rest, the vortex shedding frequency, $f_{st}$, at a certain wind speed can be expressed as the non-dimensional vortex shedding frequency based on the diffuser height, which is termed the Strouhal number and is defined as follows:

$$St = \frac{f_{st}H}{U_0} \qquad (3)$$

Once the Strouhal number, $St$, for the diffuser cross-section at rest is determined, Eq. (3) enables the critical wind speeds to be estimated by substituting $f_{st}$ with the natural frequencies presented in Table 1.

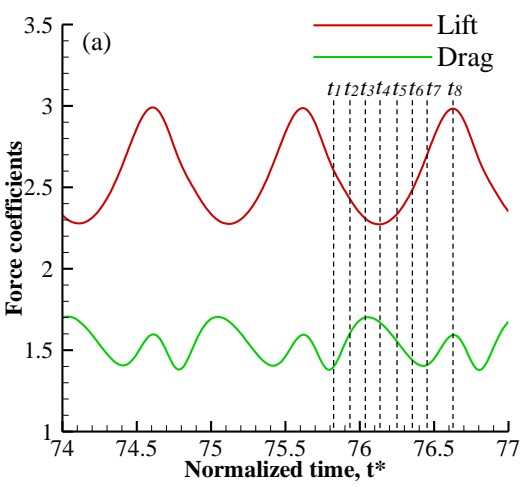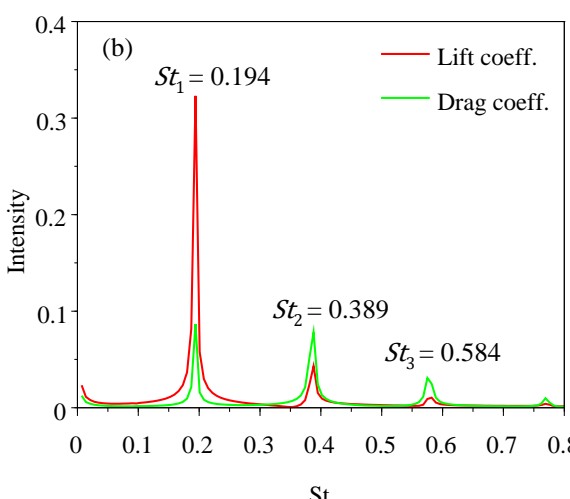

**Figure 10. The aerodynamic forces acting on the diffuser at rest: (a) The time history of the lift and drag coefficients, and (b) the frequency spectrum of the lift and drag coefficients**

The fluctuation of the lift and drag coefficients over time is shown in Fig. 10a. The mean value of the lift coefficient is 2.60, and the root mean square of its amplitude is 0.249. The mean value of the drag coefficient is 1.55, and the root mean square of its amplitude is 0.104. The positive lift acting on the wind lens means the flow speed inside the diffuser is always higher than that outside. Both forces in Fig. 10a include high harmonic wave components, which imply that multiple modes of the periodic vortex shedding affect the pressure of the wind lens. The frequency spectrum of the lift and drag coefficients

converted from their time histories by using Fast Fourier Transform (FFT) is shown in Fig. 10b. The three distinct peaks indicate the existence of three vortex modes, from which the Strouhal numbers were able to be determined. The vortex mode that appeared at the lowest frequency was defined as the first Strouhal number, $St_1$, which is 0.194. Under the identical



Reynolds number condition, the Strouhal number of 0.194 for the wind lens is slightly smaller than that for the cylinder, which is 0.2016 (Norberg, 2003), and larger than that for the square cylinder, which is 0.147 (Bai and Alam, 2018). The

second Strouhal number for the wind lens, $St_2$, is 0.389, which is close to double that of $St_1$. The third Strouhal number is 0.584. In summary, the multiple vortex modes related to both lift and drag forces appeared around the wind lens model, and the Strouhal numbers are basically integer multiples of $St_1$. Each vortex mode is associated with lift and drag forces, where the first vortex mode is primarily related to the lift, and the second and third vortex modes are primarily related to the drag.

**Figure 11. The vortex shedding behind the cross-section of the rigid wind lens model at rest during one cycle with the pressure vectors marked ($t_1$~$t_8$)**



The process of the vortices generated around the brimmed diffuser is demonstrated in Fig. 11, where the contour indicates the positive Q-criterion. The normalized times, $t^* = t_1$ to $t_8$, corresponds to the graph in Fig. 10a. The arrows marked along the wind lens cross-section are the pressure vectors normal to the diffuser surface. The drag force is chiefly attributed to the pressure acting on the brim, and the lift force acts on the curved line of the brimmed diffuser. At $t^* = t_1$, a clockwise vortex, $V_1^-$, starts shedding at the brim tip, the vortex develops on the back surface of the brim as time elapses ($t^* = t_1$–$t_4$). Accordingly, the drag increases with the development of $V_1^-$ and reaches the maximum at $t^* = t_3$. Then, the vortex is finally detached and recedes from the brim surface at $t^* = t_5$ to $t_8$. The clockwise vortex, $V_1^-$, induces the downward flow behind the brim, which generates the secondary vortex, $V_2^-$, at the bottom edge of the brim at $t^* = t_1$ to $t_4$. At the same time, the counter vortex, $V_2^+$, is accompanied under $V_2^-$. These secondary vortex pairs, $V_2^+$ and $V_2^-$, induce backflow on the bottom surface behind the throat, and the separation point of the mainstream on the bottom surface moves toward the front of the diffuser. At $t^* = t_4$, when the flow separation point reaches a point where it moves as far forward as possible, the lift becomes a minimum as shown in Fig. 10a. At $t^* = t_5$, while $V_2^-$ diminishes, $V_2^+$ does not disappear but grows larger, which needs to be treated differently as another vortex, $V_1^+$. The vortices, $V_1^+$ and $V_1^-$, form a primary vortex pair. At $t^* = t_6$–$t_8$, the $V_1^+$ travels upward near the back surface of the brim, developing. When $V_1^+$ approaches the back surface of the brim at $t^* = t_8$, the drag increases to the local maximum as shown in Fig. 10a. At this moment, the maximum lift is produced because the flow inside the diffuser is attracted by the growing $V_1^+$ and is accelerated. The first and second vortex frequencies shown in Fig. 10b are mainly attributed to the primary and secondary vortex pairs, respectively.

Based on the Strouhal numbers determined from the multiple vortex modes, it is possible to calculate the vortex frequencies with respect to wind speeds, and the critical wind speed can be estimated by comparing the natural frequencies computed in the 3D modal analysis.





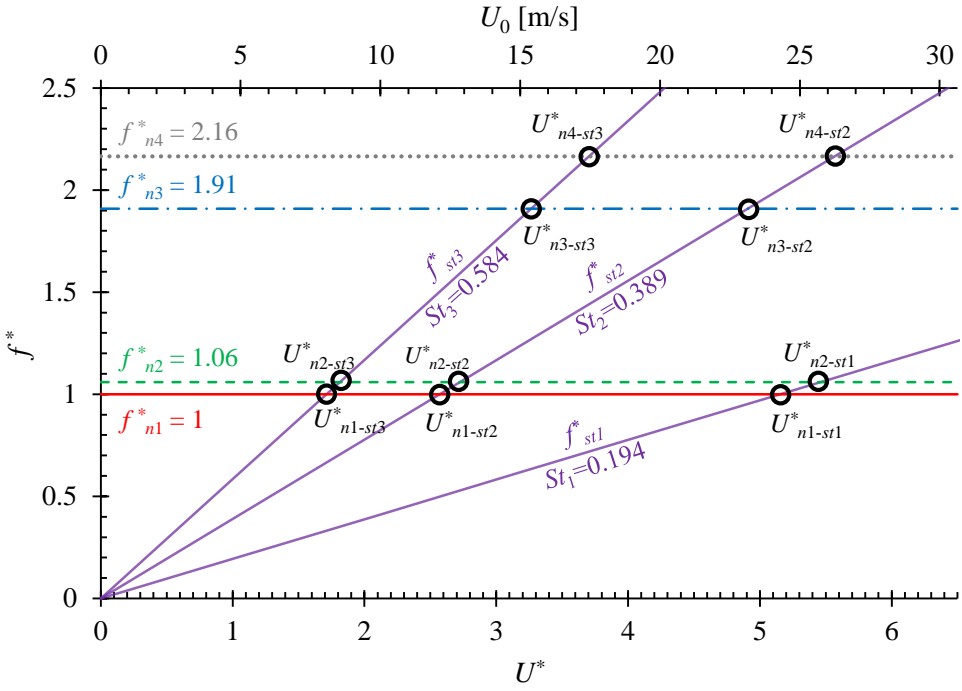

**Figure 12. A Campbell diagram based on the vortex shedding frequencies calculated by the three Strouhal numbers and the natural frequencies of the wind lens**

**Table 3. The estimated critical wind speed from the Campbell diagram**

|  | 1st vortex mode | 2nd vortex mode | 3rd vortex mode |
|---|---|---|---|
|  | ($f^*_{st1}$, $St_1 = 0.194$) | ($f^*_{st2}$, $St_2 = 0.389$) | ($f^*_{st3}$, $St_3 = 0.584$) |
| **1st natural mode** ($f^*_{n1} = 1.00$) | $U^*_{n1\text{-}st1} = 5.15$ $U_{0,\,n1\text{-}st1} = 24.3$ m/s | $U^*_{n1\text{-}st2} = 2.57$ $U_{0,\,n1\text{-}st2} = 12.1$ m/s | $U^*_{n1\text{-}st3} = 1.71$ $U_{0,\,n1\text{-}st3} = 8.1$ m/s |
| **2nd natural mode** ($f^*_{n2} = 1.06$) | $U^*_{n2\text{-}st1} = 5.46$ $U_{0,\,n2\text{-}st1} = 25.8$ m/s | $U^*_{n2\text{-}st2} = 2.72$ $U_{0,\,n2\text{-}st2} = 12.9$ m/s | $U^*_{n2\text{-}st3} = 1.81$ $U_{0,\,n2\text{-}st3} = 8.6$ m/s |
| **3rd natural mode** ($f^*_{n3} = 1.91$) | $U^*_{n3\text{-}st1} = 9.84$ $U_{0,\,n3\text{-}st1} = 46.4$ m/s | $U^*_{n3\text{-}st2} = 4.91$ $U_{0,\,n3\text{-}st2} = 23.1$ m/s | $U^*_{n3\text{-}st3} = 3.27$ $U_{0,\,n3\text{-}st3} = 15.4$ m/s |
| **4th natural mode** ($f^*_{n4} = 2.16$) | $U^*_{n4\text{-}st1} = 11.16$ $U_{0,\,n4\text{-}st1} = 52.7$ m/s | $U^*_{n4\text{-}st2} = 5.57$ $U_{0,\,n4\text{-}st2} = 26.3$ m/s | $U^*_{n4\text{-}st3} = 3.70$ $U_{0,\,n4\text{-}st3} = 17.5$ m/s |

The diagram plotted in Fig. 12 is called a Campbell diagram (Campbell, 1924), which is used to estimate resonance speed (Ewins, 2010). The reduced velocity, $U^* = U_0/(f_{n1}H)$, on the bottom horizontal axis ranges from 0 to 6.5, which correspond to 0–30.67 m/s in the actual 3 kW wind lens on the top horizontal axis. The vertical axis is the normalized frequency, $f^*$, which

represents a frequency divided by the first natural frequency, $f_{n1}$. The other $k$-th natural frequencies are expressed as $f_{nk}$. The three linear slopes shown in Fig. 12 are the three vortex shedding frequencies of the diffuser at rest calculated from Eq. (3)





for each Strouhal number. The four horizontal lines represent the natural frequencies of the wind lens. As shown in Fig. 12, there are ten intersections between the vortex shedding frequencies and the natural frequencies, which indicate possible critical wind speeds. For instance, the first vortex mode crosses the first and second natural frequencies at $U^* = 5.15$ and $5.46$,

which are expressed as $U^*_{n1-st1}$ and $U^*_{n2-st1}$, respectively. The other critical wind speeds are presented in Table 3. The severity of the resonance vibration depends on the relationship between the mode shape and the aerodynamic pressure distribution, which cannot be predicted with the Campbell diagram. Additionally, a noticeable vibration might not occur at the estimated critical wind speed in the case where a vortex mode does not have enough intensity to cause resonance in the structure, or, a severe vibration might be produced at a speed differing from the estimated critical wind speed if an aeroelastic phenomena other than the VIV occurs, such as galloping or flutter.


### 4.3 Aeroelastic response

Prior to the aeroelastic simulation, an investigation was conducted on how much the aerodynamic pressure distribution of the vortex shedding affects the four natural modes. The time-variant generalized force, $Q_k$ in Eq. (1), exerted on each mode was calculated on the condition that the wind lens section is at rest, which indicates a correlation between the aerodynamic

pressure distribution (the pressure vectors marked in Fig. 11) and each mode shape. In this paper, $Q_k$ was evaluated with respect to its intensity in the frequency domain. Each mode shape was normalized such that the vector norm was unity.

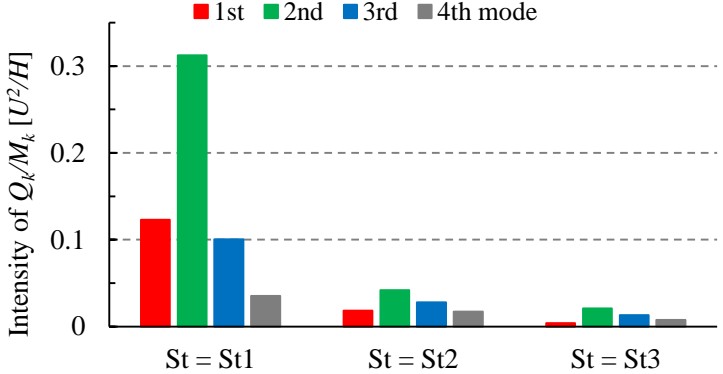

**Figure 13. The peak intensity of the ratio of generalized force to mass calculated by the three vortex shedding frequencies in the case where the wind lens is at rest**

The peak intensities extracted from the frequency spectrum of the generalized force for the four natural modes at each Strouhal number are shown in Fig. 13. $Q_k/M_k$ on the vertical axis indicates how much each vortex mode is capable of exciting vibration with each natural mode under the identical natural frequency condition. It is apparent that the second natural mode, or the rotational mode, is most susceptible to the respective vortex modes among the four natural modes. The peak intensities for the other three natural modes are comparable with each other at the respective Strouhal numbers.



However, the first natural mode, or the radial mode, is more susceptible to the first vortex mode than the second and third

vortex modes because the first vortex mode is primarily associated with the lift force acting in the radial direction.

The aeroelastic response of the wind lens model to the vortex shedding was simulated at wind speeds of 1 to 30 m/s at an

interval of 1 m/s, which correspond to the reduced flow velocities ranging from 0 to 6.36, and with the Reynolds number

fixed at 288. First, the responses of the generalized coordinates of the four modes were investigated at each flow velocity.

The generalized coordinate expressed how much each mode was excited by the external force at a certain velocity. In general,

the wind load is proportional to the square of the wind speed. To remove the increase of the wind load with the wind speed

increasing, the non-dimensional compliance, $C_k^*$, is defined as the ratio of the $k$-th generalized coordinate to the wind load,

given by,

$$C_k^* = \frac{(q_k - q_{k,\mathrm{avg}})_{\mathrm{RMS}}}{HU^{*2}} \qquad (4)$$

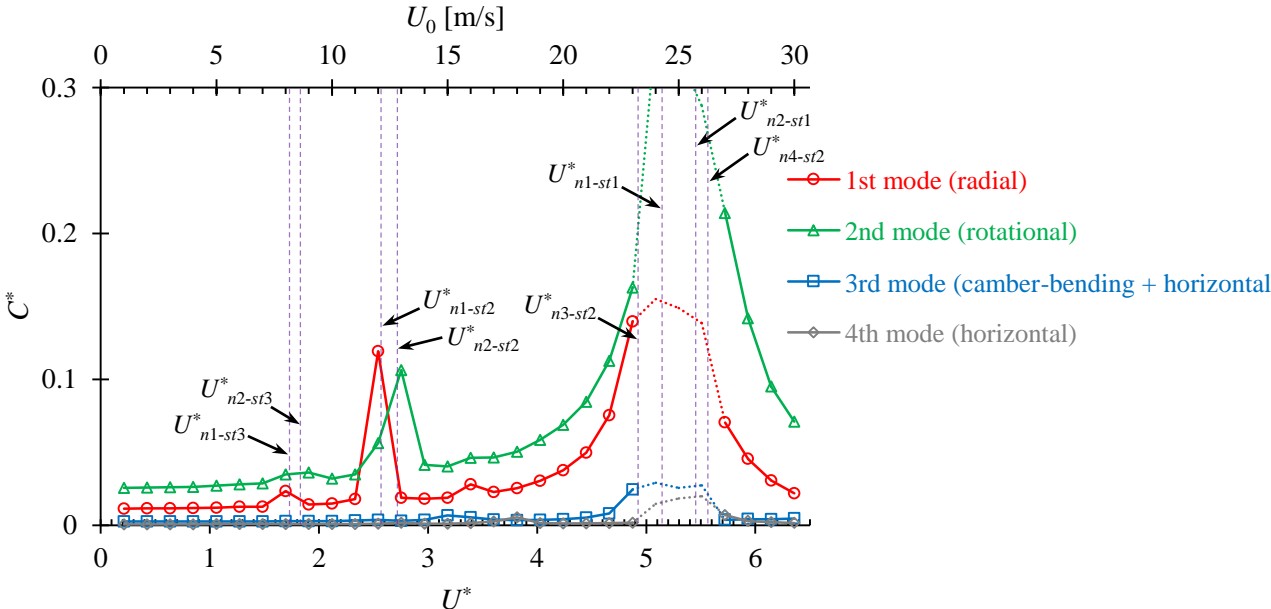

**Figure 14. Compliance of each mode with respect to wind speed in the aeroelastic analysis**

The non-dimensional compliances of the four modes with respect to the wind speed are shown in Fig. 14, with the

estimated critical wind speeds marked. The second natural mode is the most dominant of the four natural modes throughout

the flow velocity, and the first natural mode also tends to be easily affected by the vortex shedding. Therefore, the responses

of the first and second natural modes agree with their intensities of the generalized force, as presented in Fig. 13. The dotted

lines of the generalized coordinates between $U_0 = 23$ and 27 m/s ($U^* = 4.87$ and 5.72) in Fig. 14 signify the calculations were

forcedly suspended due to the extremely large displacement, which exceeds the limitation of the grid system in the CFD





program. The values on the dotted lines were temporarily extracted from a few cycles until the calculations ceased. Even though the compliances have not been calculated in a steady oscillation between the two wind speeds, it is reasonable to infer that the highest peak of the first and second natural modes would appear between $U_0 = 24$ and 26 m/s ($U^* = 5.09$ and

5.51) in Fig. 14. These wind speeds are close to the critical wind speeds of $U_{0,\,n1\text{-}st1}$ and $U_{0,\,n2\text{-}st1}$, which are evaluated at the intersections between the first and second natural frequencies and the first vortex frequency shown in Fig. 12 and Table 3. The sharp increase in both the first and second compliances near 24 m/s is possibly caused by resonance, in which the first vortex frequency synchronizes with either or both the first or second natural frequencies. There are other distinct peaks in the first and second natural modes at 12 and 13 m/s ($U^* = 2.54$ and 2.76), respectively, which are also close to the estimated

critical wind speeds of $U_{0,\,n1\text{-}st2}$ and $U_{0,\,n2\text{-}st2}$. Similarly, there is a small rise in the third and fourth modes near $U_{0,\,n3\text{-}st2}$ and $U_{0,\,n4\text{-}st2}$, respectively, which was the result of the second vortex mode synchronizing with the third and fourth natural frequencies. However, the peak compliances at the critical wind speeds of the second vortex mode are low because the intensity of the second vortex mode is weaker than that of the first one, as shown in Fig. 10b. The third vortex mode does not have a prominent effect on the vibration excitation of the wind lens structure, considering that there are only tiny peaks of

the compliances near $U_{0,\,n1\text{-}st3}$ and $U_{0,\,n2\text{-}st3}$.

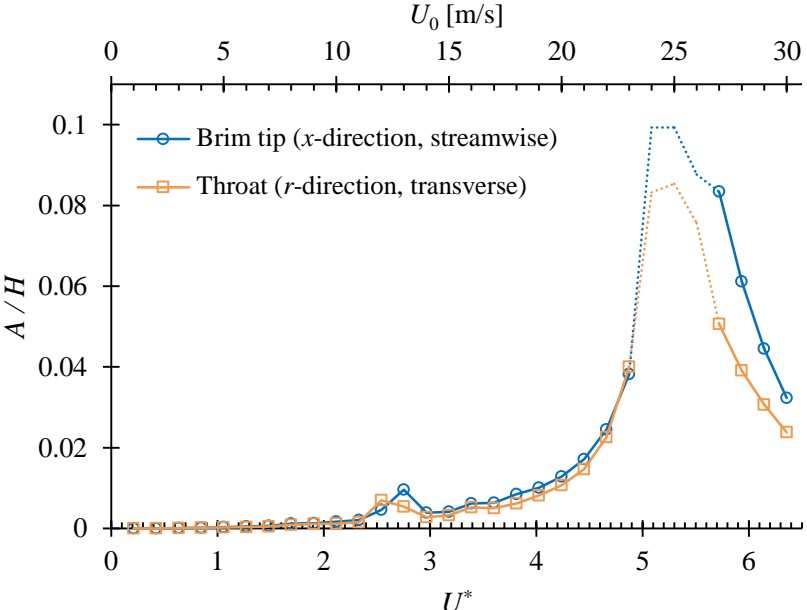

**Figure 15. The displacement amplitude of the brim tip in the *x*-direction and the throat in the *r*-direction**

    The displacement amplitude of the diffuser with respect to the wind speed and the reduced velocity is shown in Fig. 15. The vertical axis expresses the ratio of the displacement amplitude, $A$, to the diffuser height, $H$. This study focused on the

displacement amplitude of two representative locations of the diffuser: (1) the streamwise ($x$) motion at the brim tip, and (2)





the transverse ($r$) motion at the throat. The first one is associated with the rotational, horizontal, brim-bending, and camber-bending modes and the second one is associated with the radial, rotational, and camber-bending. As shown in Fig. 14, the third and fourth natural modes are excited by the weak second vortex mode only, while the responses of the first and second natural modes are noticeable because the radial and rotational modes are more sensitive to the three vortex modes. Thus, the

displacement amplitudes shown in Fig. 15 are mostly attributed to the radial and rotational modes. Also, in Fig. 15, the dotted lines between and 27 m/s in the graph represent where the calculations were aborted. Even if stable oscillations were not obtained, it is obvious that the largest $A/H$ at the brim tip and the throat would appear between 24 and 26 m/s. The similar displacement amplitudes at both locations indicate that the resonance pattern of the wind lens section is principally the rotational mode, which accords with the result of the generalized force shown in Fig. 13. There are the other peaks on the

curve of the brim tip at 13 m/s and on that of the throat at 12 m/s. These peaks indicate that the second vortex mode is also influential enough to induce the radial and rotational modes as with the first vortex mode. However, the effect of the third vortex mode is barely identifiable in Fig. 15. The peak value of the brim tip at 13 m/s is 0.00896 ($A = 3.87$ mm), and that of the throat at 12 m/s is 0.00679 ($A = 2.93$ mm). In consideration of the 1 mm thick diffuser shroud, the displacement amplitudes of 2.93 mm and 3.87 mm are substantial. Since these critical wind speeds calculated in this simulation are close

to the average wind speed of 10.6 m/s on April 3, 2012, the real VIV in the wind lens that was observed could have been caused by the second vortex mode. Most importantly, these critical wind speeds are in good agreement with the estimated critical wind speed from the Campbell diagram presented in Fig. 12. In practice, the WLTs are usually operated in the turbulent flow regime, where the Reynolds number is much higher than 288, and the Strouhal number varies with the Reynolds number. Nevertheless, the estimated critical wind speeds are reasonable. In the case of the circular cylinder, the

Strouhal number ranges from approximately 0.18 to 0.22 at $200 < Re < 100,000$ (Blevins, 1990). The Strouhal number for the square prism has a maximum of 0.16 at $Re \sim 200$, and it varies from 0.12 to 0.13 at $Re > 1,000$ (Bai and Alam, 2018). As mentioned in Sect. 4.2, the Strouhal number for the wind lens is 0.194 at a Reynolds number of 288; however, the Strouhal numbers for the wind lens section depending on the Reynolds number have not been sufficiently studied yet. Unless the Strouhal numbers change drastically, estimation of the critical wind speeds using the Campbell diagram is appropriate.

Furthermore, the radial and rotational mode is a degenerate mode, which vibrates partially in the diffuser, as explained in Sect. 4.1. This fact indicates that the VIV of the wind lens, which is caused by the radial and rotational modes, may not occur in the entire diffuser structure but a local section of the diffuser. If so, the 2D aeroelastic simulation in this study is valid to predict a trigger of such a local VIV and to approximately estimate the critical wind speed, although a precise quantitative estimation is not provided.

Finally, in order to investigate the influence of the vortex shedding frequencies on the aeroelastic analysis, the frequency spectrum was calculated from the time-variant lift and drag at each wind speed in the FFT analysis. The intensities of the lift and drag spectra with respect to the reduced velocity are expressed as a contour map in Fig. 16. As mentioned, the zone between 23 to 27 m/s, where the computations were aborted due to the excessive deformation, is blank. The straight ridgelines of the contours completely coincide with the vortex-frequency lines estimated from the Strouhal number shown in



the Campbell diagram (Fig. 12), which indicates that the Strouhal number is not affected by the aeroelastic vibration, except in the range over 27 m/s, where another horizontal ridgeline appears between the first and second natural frequencies. This fact implies that a lock-in phenomenon occurs at those wind velocities when the large vibration is excited by the first vortex mode. On the other hand, there is no lock-in near the intersection between the second vortex mode and the four natural modes because of the comparatively-small-amplitude vibration.


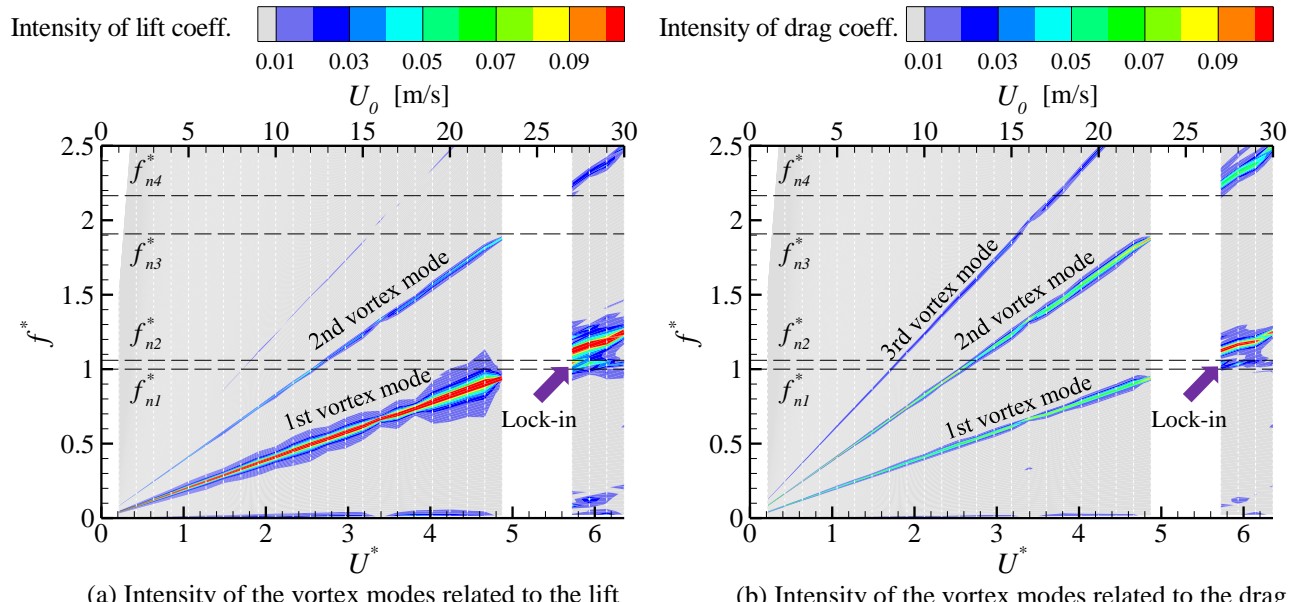

(a) Intensity of the vortex modes related to the lift          (b) Intensity of the vortex modes related to the drag

**Figure 16. Surface contour based on the frequency spectrum of lift and drag at each wind speed in the aeroelastic analysis**

## 5 Conclusions

The aeroelastic simulation of the FIV for the brimmed diffuser section was performed using 3D FEM and 2D CFD at the

wind speeds ranging from 0 to 30 m/s ($U^* = 0$ to 6.36) under the condition of the Reynolds number of 288. The numerical results concluded that the self-induced vibration observed in the actual wind lens structure is VIV. Neither galloping nor flutter occurred in the wind lens structure because the five support arms restrained the wind lens from changes in the angle of attack, as well as prevented a large displacement. Nevertheless, the structural vibration occurred in the brimmed diffuser due to the mode shapes critical to VIV: the radial and rotational modes. From the results of the 3D modal analysis, the

vibrational modes in the diffuser cross-section can be classified into five basic patterns: radial, rotational, horizontal, camber-bending, and brim-bending modes. The radial and rotational modes are subjected to the aerodynamic pressure due to the vortex shedding modes, compared to the other natural modes. The two critical modes, which are accompanied by the circumferential bending oscillation of the support arms, are difficult to prevent because of the limitation of the aerodynamic





design and the diffuser mass. The lift and drag acting on the brimmed diffuser have high harmonic oscillations due to multiple vortex modes, which are associated with vortices shed from the brimmed diffuser. In the aeroelastic analysis, the response of the elastic brimmed diffuser has peak amplitudes in the displacement at the critical wind speeds estimated from the Campbell diagram, which is based on the Strouhal numbers calculated in the case where the brimmed diffuser is at rest. The two critical modes are excited by the first vortex mode at the wind speed of 24 to 27 m/s ($U^*$ = 4.87 to 5.72), which results in considerably large oscillation. Also, the second vortex mode excites the radial and rotational modes at 12 and 13 m/s ($U^*$ = 2.54 to 2.76), respectively, which causes noticeable oscillation with a relatively small amplitude. These wind speeds of 12 and 13 m/s are close to the average wind speed when the actual VIV of the brimmed diffuser was observed. Although the 2D aeroelastic analysis neglects some complex phenomena that occur in the actual wind lens, it provides a reasonable estimation of the critical wind speeds for VIV in the wind lens structure.

**Appendix A: Time-step and mesh dependency check**

Appendix A demonstrates the convergence of the numerical results with different time steps and the meshes in the CFD analysis explained in Sect. 3.2. The convergence was verified by comparing each frequency spectrum of the lift and drag coefficients. Figures A1 and A2 show the intensity of lift and drag in the frequency domain with the different time steps and with different grid systems, respectively. In Fig. A1, the numerical result with half the current time step is compared to that with the current time step. In Fig. A2, the numerical results in the grid systems with 0.66, 0.77, 0.88, and 1.13 times the current elements are compared to that with current elements. As seen in Fig. A1, the peaks occur at the same Strouhal numbers in both time steps, except for the third Strouhal number in Fig. A1b. It was confirmed that the small error in the third Strouhal number from the drag is inconsequential in the result, which is irrelevant as the third vortex mode is inconsequential in this study. Although there are small differences in the peak widths, the current time step was used in this study because computational time can be saved considerably, and the differences are too small to have any influence on the results. From Fig. A2, it is evident that the solution is converged as the number of elements increase. There are small peaks at $St = 0.5St_1$, $1.5St_1$, and $2.5St_1$ in the grid system with fewer elements; however, they disappear when the number of elements exceeds 0.88 times the current elements. The reason why the grid system with 732,451 elements was chosen for this study is that the grid system with 0.88 times the current elements still has a small risk of the appearance of unnecessary peaks, and the grid system with 1.13 times the current elements requires a longer computational time.



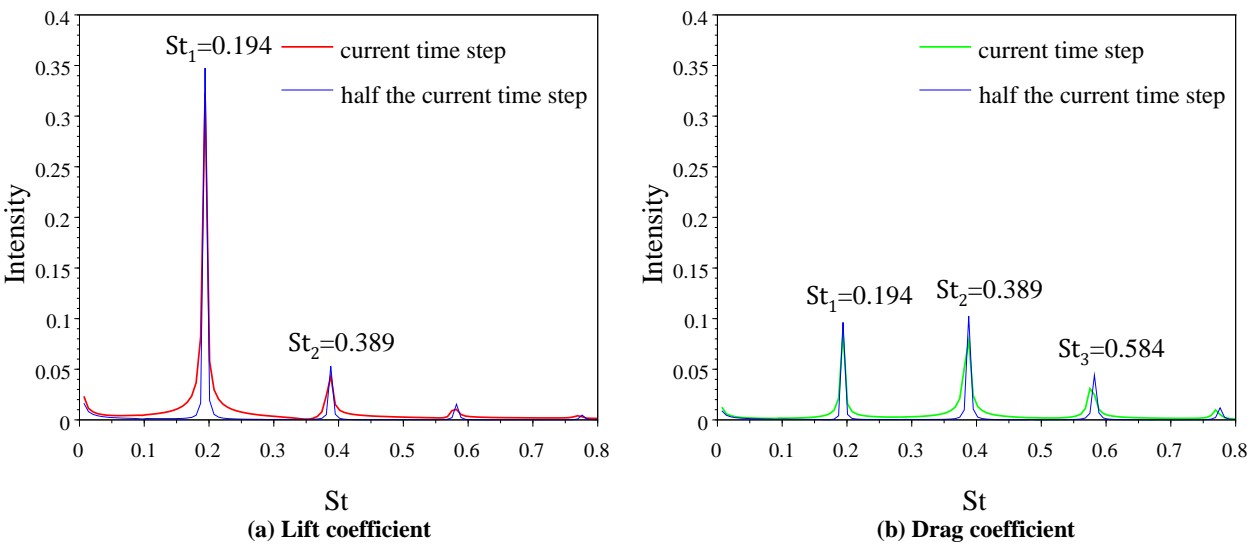

**Figure A 1. The frequency spectra of the force coefficients in two different time steps in the current grid system (732,451 elements)**

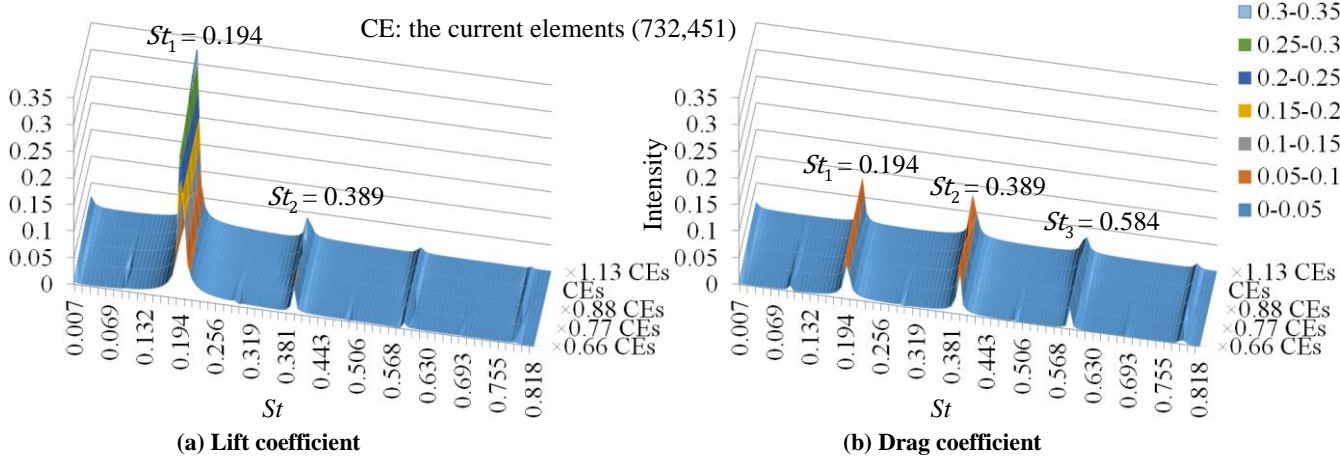

**Figure A 2. The frequency spectra of the force coefficients in the current time step in six different grid systems**

Author contributions. TK performed the analyses, compiled the literature, and wrote the corresponding paper. HN developed the 2D aeroelastic CFD program and performed the analyses. NU guided the current study and helped write the corresponding paper as TK's supervisor. OY, the wind lens developer, provided the information and data of brimmed diffusers.



Competing interests. The authors declare that they have no conflict of interest.

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
