# Peer review of "Fundamental effect of vibrational mode on vortex-induced vibration in a brimmed diffuser for a wind turbine"

_Wind Energy Science, 2020_

## Referee Comment (RC1) · Anonymous Referee #1 · 10 May 2020

**Referee report WES-2020-29**

May 10, 2020

**1 Summary**

The authors present an aeroelastic simulation of a brimmed diffuser on a small wind turbine (3 kW). The brimmed diffuser is referred to as a wind lens. In some circumstances the brimmed wind turbine is known to be subject to vibrations. The authors attribute this to the excitation of eigenmodes by vortices shed periodically from the brim and perform an aeroelastic analysis of the system. First, modal analysis is performed based on a finite-element simulation. Then, low Re CFD simulations are performed on the wind lens and the vortex shedding frequency (a function of wind speed) is compared to the eigenfrequencies of the system and the aeroelastic response is investigated.

**2 Interest**

The paper addresses a real but rather specific problem, with limited general interest. No attempt is made to generalise the conclusions to other similar systems.

**3 Low Reynolds approach**

The authors have chosen to perform their CFD analysis at Re = 288 (by scaling the viscosity) rather than the Re = 3e5 corresponding to the real operating conditions. The reason given is "to diminish the instability of turbulence". The authors should explain exactly what they mean by this. The authors do not explain why they believe it is warranted to run the simulations at such a low Reynolds number, apart from a cursory reference to the original and viscosity-scaled models having the same mass ratio. This

is a major shortcoming of the manuscript, leading me not to recommend publication in its present form. I see at least the following issues:

- Can one assume that the Strouhal number is roughly the same at Re = 288 and Re = 3e5? It appears to me that this assumption is implied. The discussion in 4.3, though certainly useful, does not demonstrate that this assumption is justified.

- At 3e5, the vortex shedding is expected to be 3D. How can this not affect the excitation?

This should be addressed. Calculations at higher Re should be seriously considered.

**4   Rotor**

The rotor appears to be completely absent from the calculations. This is obviously computationally expedient but possibly questionable. The rotor will at the very least cause the wind speed at the rotor to be lower than the incoming wind. How was this taken into account? Are tip vortices from the blades expected to interact with the vortices shed by the brim of the diffuser? I assume it was verified that the rotor cannot excite the diffuser, but it should be mentioned.

The authors should mention explicitly that the rotor was left out, why they believe this can be done or had to be done and why a low-fidelity approach (actuator disk and related models) was not considered. I understand that a full rotor simulation is outside the scope of the manuscript.

**5   Modal analysis**

The value of the paper could be improved with experimental or operational modal analysis, as the authors appear to have access to a physical setup. This is not difficult to do and would provide a very useful validation of the numerical modal analysis.

The authors should seriously consider experimental or operational modal analysis.

**6   Estimation of the critical wind speeds**

I disagree with the statement that the presence of harmonics points to the presence of three separate vortex modes. Harmonics will appear in the FFT

from the moment the variation the signal is not a perfectly sinusoidal function. A single mode is therefore perfectly possible. There also appears to be little basis for the statement the first purported mode is primarily related to lift and the others to drag. Every vortex shedding mode will have a signature in both the lift and drag forces, even though obviously the period of drag variations is only half the period of the lift variations. To avoid misunderstanding: I do agree of course that the harmonics play a role in the interaction with the eigenfrequencies.

This should be corrected. If the authors believe I am mistaken, they are welcome to demonstrate the presence of truly separate modes.

**7 Lock-in**

The lock-in phenomenon deserves a more thorough discussion. Its identification in fig. 16 is not convincing. For a formal definition of lock-in, the authors may refer to Kumar, Navrose, and Mittal, Physics of Fluids 28 (2016) [doi: 10.1063/1.4967729] .

I recommend that the authors argue the presence of lock-in better.

**8 Writing and figures**

The writing is clear and almost without grammatical errors. The figures are clear.

**9 Minor comments**

- The authors use a compressible solver in the CFD. Why do they expect compressibility to be important?

- Two letter symbols such as St and Re should not be italic

---

## Referee Comment (RC2) · Anonymous Referee #2 · 9 Jun 2020

The paper proposes an interesting study on the aero-elastic response of a wind lens, used as a diffuser for a ducted horizontal-axis wind turbine. The topic is relevant for the journal and considers an interesting topic for researchers and technicians involved in the design of ducted wind turbines. Moreover, the study involves a combined aero-structural time-resolved analysis of a realistic wind lens, proposing an approach that might be used in future studies for the design of such devices, that have the potential for improving the wind turbine energy harvesting. The paper is well structured, well written, and the figures are easily readable and of good quality (the introduction might be probably written in a more concise and effective way).

Despite these general positive aspects of the paper, the referee is however surprised by the way in which the authors carried out the CFD analyses reported in the paper, which introduces a very relevant issue in the paper quality and technical relevance. The major points of discussion on this aspect are reported below:

1 Authors decided to focus on 2D simulations, which is reasonable due to the high computational cost of aero-structural simulations, however more specific justification on the technical relevance of this choice is required.

2. Authors make use of a compressible-flow model. Since the Mach number should be well below 0.3, this choice does not seem justified and might even lead to numerical issues if proper preconditioners are not used. Please clarify in the text.

3. The authors use a deforming mesh strategy to account for the deformation of the structure during motion; some information on the numerical technique used for deforming the mesh should be reported (at least with a citation). Did the Authors experience any issue with cell quality due to mesh deformation during the simulations?

4. No model of the turbine within the wind lens is considered, at least a comment on this aspect should be reported.

4. Authors decided to reduce by three orders of magnitude the Reynolds number of the problem, altering the flow regime from turbulent (the actual one) to laminar. Authors state that similarity in aero-elastic response is confirmed despite the alteration of the Reynolds number, but they do not explained how and why. As well known, turbulence is a non-linear phenomenon which does not only feature unsteadiness and instability, but which alters the gross properties of the flow and, hence, the values of lift and drag coefficients, as well as the vortex shedding frequency. Why do the Authors believe that such an arbitrary change in flow regime does not lead to unrealistic results?

This last aspect is by far the most critical of the paper. As well highlighted by Figure 11, the aerodynamic forcing acting on the wind lens is determined by the detachment of

vortices connected to flow separations in different areas of the wind lens profile. Such a phenomenon is dominated by viscous/turbulent effect and there is no proof that the shedding frequencies, which ultimately allow constructing the Campbell diagram in Figure 12, are estimated in a realistic way by the CFD model. Authors are clearly aware of this and discuss the issue in page 19, without however giving a convincing explanation of the validity of their approach. In this context, it is not even clear the motivation for such a severe simplification: with present-day CFD technology it is possible to simulate highly turbulent flows with U-RANS, employing suitable turbulence models (such Spalart-Allmaras or k-w SST models). Such models might be not entirely reliable when dealing with separated flows and, in general, aerodynamic instabilities, however they were shown to predict correctly the Strouhal number of vortex shedding; this referee believes that the uncertainty is introduced by altering the flow regime from turbulent to laminar is much higher than that introduced by the use of a turbulence model.

As a final consideration, this referee believes that this paper documents a general aero-structural methodology which is scientifically interesting, but featuring a very critical technical issue on the aerodynamic side which ultimately reduces the technical relevance of the entire work. Before the paper is considered for publication, a revision is required in which the Authors justify better their choices and demonstrate, with at least a U-RANS simulation with a rigid wind lens, that the vortex shedding frequencies for the realistic Re of 300000 are nearly the same as those predicted for Re = 288.

---

## Editor Comment (EC1) · Alessandro Bianchini (Editor) · 10 Jun 2020

Dear authors, the Reviewers pointed out some significant criticalities in your work. Please carefully analyze their comments and post at your earliest convenience detailed answers to them. After the answers will be in, I will take my decision on the paper. Best regards

---

## Author Comment (AC1) · 15 Jul 2020

**Authors' Responses to the comments of Referee #1 on**

"Fundamental effect of vibrational mode on vortex-induced vibration in a brimmed diffuser for a wind turbine" by Taeyoung Kim et al. (wes-2020-29)

We really appreciate your valuable comments to improve the quality of this manuscript. Our responses to your comments are as below.

**3 Low Reynolds approach**

**Comment**

"The authors have chosen to perform their CFD analysis at Re = 288 (by scaling the viscosity) rather than the Re = 3e5 corresponding to the real operating conditions. The reason given is "to diminish the instability of turbulence". The authors should explain exactly what they mean by this."

**Response**

As mentioned in Sect. 1, the instability of turbulence in this context means disorganized wakes, 3D vortex patterns, and the surface condition at the critical Reynolds number regime. As the reviewer pointed out, this word is confusing because we did not simulate the actual VIV in the wind lens. For this reason, we plan to rewrite these confusing words as below.

**Revision**

**Line 168–171, Page 7:** The corresponding sentence, "As explained in Sect. 1, … in the 2D simulation.", will be replaced as follows.

As explained in Sect. 1, to minimize the complexity caused by the actual high Reynolds number and to elicit the fundamental mechanism of VIV, the Reynolds number of the current brimmed diffuser model was set to 288 by modifying the dynamic viscosity in the 2D numerical simulation.

**Comment**

"The authors do not explain why they believe it is warranted to run the simulations at such a low Reynolds number, apart from a cursory reference to the original and viscosity-scaled models having the same mass ratio. This is a major shortcoming of the manuscript, leading me not to recommend publication in its present form. I see at least the following issues:

Can one assume that the Strouhal number is roughly the same at Re = 288 and Re = 3e5? It appears to me that this assumption is implied. The discussion in 4.3, though certainly useful, does not demonstrate that this assumption is justified.

At 3e5, the vortex shedding is expected to be 3D. How can this not affect the excitation?

This should be addressed. Calculations at higher Re should be seriously considered."

**Response 1**

This study intends to explain and clarify numerically how the vortex-induced vibration (VIV) occurs in the wind lens structure rather than to perform the realistic simulation of the flow-induced vibration (FIV) of the actual wind lens. Unintendedly, the introduction of the manuscript might lead readers to misunderstand the object of this study, as the reviewer pointed out. We plan to add supplementary explanations to the introduction of the revised manuscript to emphasize the novelty of this paper, including the reason why we reduced our numerical model to the two-dimension and the low Reynolds number, as follows.

**Revision**

**The 1st paragraph on Page 4:** The sentence, "Motivated by … numerical simulation", is replaced to the following

Despite many previous studies of VIV for bluff bodies, there is little fundamental understanding of VIV of the brimmed diffuser, whose aerodynamic characteristics contain those of both bluff bodies and airfoils. The brimmed-diffuser section always generates a larger mean lift force with circulation than a mean drag force. On the contrary, the general bluff bodies, such as a circular or rectangular cylinder, generate lift and drag fluctuation, in which the lift force averages zero over time. From this point of view, the aerodynamics of the brimmed-diffuser section resembles that of an airfoil with a large flap angle at a high angle of attack after stall, rather than that of the bluff bodies. Recently, some studies of VIV for a simple airfoil without a flap (Skrzypiński et al., 2014; Benner et al., 2019) have been conducted with the interest of VIV for a wind turbine blade in a standstill condition. Nonetheless, the VIV of an airfoil with a large flap after stall, which is similar to the brimmed-diffuser shape, has not received attention because it does not happen in general operation for aircraft and wind turbines.

Motivated by the VIV observed in the actual wind lens, we numerically investigated the fundamental mechanism of VIV for the brimmed-diffuser shape. Considering the average wind speed on April 3, 2012 and the size of the wind lens, the wind lens turbine was in operation at a Reynolds number of approximately 300,000. The Reynolds number near 300,000 is in the critical Reynolds number regime, where the laminar boundary layer of a bluff body undergoes a turbulent transition, and the wake is disorganized and has 3D vortex structures (Williamson, 1996; Anderson, 2007). Furthermore, the vortex shedding frequency is sensitive to the surface condition of the body and the turbulence intensity of the flow (Zdravkovich, 1990). Because of the complexity of the turbulence and high computational cost, the precise numerical simulation of VIV in the large Reynolds number regime is still challenging even for a circular cylinder (Sarpkaya, 2004; Nguyen and Nguyen, 2016).

Another factor to complicate the flow around the brimmed diffuser is a rotor effect. According to Hasegawa et al. (2007), the rotor inside the wind lens causes a three-dimensional flow around the wind lens, whereas the flow around the wind lens without a rotor tends to be two-dimensional. In addition, the vortex generated by the rotor blade tip induces another vortex in the boundary layer of the inner surface of the wind lens, and the induced vortex suppresses flow separation from the inner surface of the wind lens, which leads to enhancement of collection and acceleration of the wind (Takahashi et al., 2012). Then, the blade tip vortices and the induced vortices rapidly weaken with their interaction, while the blade tip vortex without the brimmed diffuser remains even in the far downstream region (Abe et al., 2009; Takahashi et al., 2012). Despite these studies about the rotor effect on the WLTs, it has been unremarked how the rotor affects the frequency or intensity of the vortex shedding from the brimmed diffuser.

To initiate the fundamental investigation of VIV for the brimmed-diffuser shape, it is reasonable to simplify the analytical model by minimizing complexities, such as the turbulent effect, 3D wake structure, and the rotor effect. Therefore, the purpose of this paper is not to simulate the observed VIV for the actual wind lens, but rather to clarify the fundamental mechanism of VIV for the brimmed-diffuser shape. To elicit the fundamental mechanism of VIV, the 2D aerodynamic model is employed at a low Reynolds number of 288 with the rotor excluded, where the vortex shedding structures can be treated as 2D in the wake. The vibrational modal characteristics for the whole 3D wind lens are calculated by using the finite element method (FEM). Coupled with the equations of motion in the modal space, 2D unsteady aeroelastic simulation based on 2D Navier-Stokes equations is performed.

**Response 2**

As mentioned in the manuscript, our results provide an approximated estimation of the critical wind speeds for the actual wind lens unless the Strouhal number changes drastically in Reynolds number, which is based on the fact that the Strouhal numbers for the circular cylinder and square prism do not change so much from $Re = 200$ to 100,000. However, the Reynolds number effect on the Strouhal numbers for the wind lens is still unknown as mentioned in the manuscript. In accordance with the reviewer's comment, we emphasize the limitation of this study in the revised manuscript, as follows:

**Revision**

**Line 383–394, Page 19:** The sentence, "In consideration … is appropriate", is replaced to the following.

The observed critical wind speed of $U^* = 2.25$ ($U_0 = 10.6$ m/s) for the actual wind lens is close to the calculated ones of $U^* = 2.54$ and 2.75 in the numerical result, which implies that the observed VIV in the actual wind lens could have been caused by the second vortex shedding frequency rather than the first vortex shedding frequency. In practice, the actual wind lens is usually operated in the turbulent flow regime at higher Reynolds numbers, but the Reynolds number effect on the Strouhal numbers of the wind lens is still unknown. As for the bluff bodies, the Strouhal number for the circular cylinder ranges from 0.18 to 0.22 at $200 < Re < 100,000$ (Blevins, 1990); the Strouhal number for the square prism has a maximum of 0.16 at $Re \approx 200$, and it varies from 0.12 to 0.13 at $Re > 1,000$ (Bai and Alam, 2018). Similarly, the Strouhal number for the wind lens can be expected to change little at $200 < Re < 100,000$. Unless the Strouhal numbers change drastically, the numerical results provide an approximate estimation of the critical wind speeds for the actual wind lens. However, according to Bai and Alam (2018), the intensities of high harmonics for the square prism are dependent on the Reynolds number. For further study about the realistic simulation of VIV in the actual wind lens, the investigations of the Reynolds number effect, a 3D flow effect, and a rotor effect are required in the future.

**Relevant minor revisions**

Additionally, we plan to delete or correct confusing sentences in the manuscript for the revised version as below to avoid the misunderstanding. Especially, the actual velocity corresponding to the reduced velocity is removed from the manuscript including Table 3, Fig. 14, Fig. 15, and Fig. 16. Even though our purpose to show both $U^*$ and $U_0$ was to help readers know how fast each reduced velocity is., we noticed that showing $U_0$ corresponding to $U^*$ confuses readers thanks to the referee's comments. Some of $U^*$ below do not correspond to the $U_0$ in the current manuscript because the mesh problem explained in Sect. 4.3 was solved and renewed reduced-velocity ranges need to be shown. The details are answered in **Response** to **7 Lock-in** at the end of this response form.

**Line 14, Page 1:** The sentence, "The 2D aeroelastic analysis provided a reasonable estimation of the critical wind speeds for the actual VIV observed in the wind lens structure", will be deleted.

**Line 15, Page 1:** The word, "Also,", will be rewritten into "As a result,".

**Line 308, Page 15:** The sentence, "which correspond to 0–30.67 m/s in the actual 3 kW wind lens on the top horizontal axis", will be deleted.

**Line 337, Page 17:** The sentence, "The aeroelastic response of the wind lens model to the vortex shedding was simulated at wind speeds of 1 to 30 m/s at an interval of 1 m/s, which correspond to the reduced flow velocities ranging from 0 to 6.36, and with the Reynolds number fixed at 288.", will be rewritten to "The aeroelastic response of the wind lens model to the vortex shedding was simulated at thirty reduced velocities ranging from 0 to 6.36 at the same intervals with the Reynolds number fixed at 288.".

**Line 350, Page 17:** between $U_0 = 23$ and 27 m/s ($U^* = 4.87$ and 5.72) $\rightarrow$ between $U^* = 5.30$ and 5.72

**Line 354, Page 18:** between $U_0 = 24$ and 26 m/s ($U^* = 5.09$ and 5.51) $\rightarrow$ near $U^* = 5.5$

**Line 355, Page 18:** $U_{0,\,n1\text{-}st1}$ and $U_{0,\,n2\text{-}st1}$ $\rightarrow$ $U^*_{,\,n1\text{-}st1}$ and $U^*_{,\,n2\text{-}st1}$

**Line 357, Page 18:** 24 m/s $\rightarrow$ $U^* = 5.09$

**Line 359, Page 18:** 12 and 13 m/s ($U^* = 2.54$ and 2.76) $\rightarrow$ $U^* = 2.54$ and $U^* = 2.76$
**Line 360, Page 18:** $U_{0,\,n1\text{-}st2}$ and $U_{0,\,n2\text{-}st2}$ $\rightarrow$ $U^*_{,\,n1\text{-}st2}$ and $U^*_{,\,n2\text{-}st2}$

**Line 360, Page 18:** $U_{0,\,n3\text{-}st2}$ and $U_{0,\,n4\text{-}st2}$ $\rightarrow$ $U^*_{,\,n3\text{-}st2}$ and $U^*_{,\,n4\text{-}st2}$
**Line 365, Page 18** $U_{0,\,n1\text{-}st3}$ and $U_{0,\,n2\text{-}st3}$ $\rightarrow$ $U^*_{,\,n1\text{-}st3}$ and $U^*_{,\,n2\text{-}st3}$

**Line 376, Page 19:** The dotted lines between $U_0 = 23$ and 27 m/s $\rightarrow$ The blank between $U^* = 5.30$ and 5.72

**Line 377, Page 19:** between 24 and 26 m/s $\rightarrow$ near $U^* = 5.5$

**Line 380, Page 19:** 13 m/s $\rightarrow$ $U^* = 2.75$

**Line 380, Page 19:** 12 m/s → $U^* = 2.54$

**Line 382, Page 19:** 13 m/s → $U^* = 2.75$

**Line 383, Page 19:** 12 m/s → $U^* = 2.54$

**Line 415, Page 20:** from 0 to 30 m/s ($U^* = 0$ to 6.36) → from $U^* = 0$ to 6.36

**Line 428, Page 21:** 24 to 27 m/s ($U_* = 4.87$ to 5.72) → $U^* = 5.30$ to 5.72

**Line 429, Page 21:** 12 and 13 m/s ($U^* = 2.54$ to 2.75) → $U^* = 2.54$ and $U^* = 2.75$

**Line 430, Page 21:** The sentence, "These wind speeds of 12 and 13 m/s are close to the average wind speed when the actual VIV of the brimmed diffuser was observed", will be deleted.

**References to be added**

Skrzypiński, W. R., Gaunaa, M., Sørensen, N., Zahle, F., and Heinz, J.: Self-induced vibration of a DU96-W-180 airfoil in stall, Wind Energy, 17, pp. 641–655, doi: 10.1002/we.1596, 2014.

Benner, B. M., Carlson, D. W., Seyed-Aghazadeh, B., and Modarres-Sadeghi, Y.: Vortex-Induced Vibration of Symmetric airfoils used in Vertical-Axis Wind Turbines, Journal of Fluids and Strucutures, 91, 102577, doi: 10.1016/j.jfluidstructs.2019.01.018, 2019.

Hasegawa, M., Ohya, Y. and Kume, H.: Numerical Studies of Flows Around a Wind Turbine Equipped with Flanged-Diffuser Shroud by Using an Actuator-Disc Model, Trans. Japan Soc. Mech. Eng. Ser. B, 73(733), 1860–1867, doi:10.1299/kikaib.73.1860, 2007. (in Japanese)

Takahashi, S., Hata, Y., Ohya, Y., Karasudani, T. and Uchida, T.: Behavior of the blade tip vortices of a wind turbine equipped with a brimmed-diffuser shroud, Energies, 5(12), 5229–5242, doi:10.3390/en5125229, 2012.

**4 Rotor**

**Comment**

"The rotor appears to be completely absent from the calculations. This is obviously computationally expedient but possibly questionable. The rotor will at the very least cause the wind speed at the rotor to be lower than the incoming wind. How was this taken into account? Are tip vortices from the blades expected to interact with the vortices shed by the brim of the diffuser? I assume it was verified that the rotor cannot excite the diffuser, but it should be mentioned. The authors should mention explicitly that the rotor was left out, why they believe this can be done or had to be done and why a low-fidelity approach (actuator disk and related models) was not considered."

**Response**

As the reviewer pointed out, the consideration of the rotor is important to a realistic full simulation of VIV for the actual wind lens. However, to diminish the aerodynamic complexity and elicit the fundamental mechanism of VIV for the brimmed diffuser, we removed the rotor from the numerical model. According to Hasegawa et al. (2007), the rotor inside the wind lens causes a three-dimensional flow around the wind lens, whereas the flow around the wind lens without a rotor tends to be two-dimensional. Thus, no-rotor model helps us discuss the aerodynamics in two-dimension. According to Takahashi et al. (2012), the vortex generated by the rotor blade tip induces another vortex in the boundary layer of the inner surface of the diffuser shroud, and the induced vortex suppresses flow separation from the inner surface of the diffuser shroud, which leads to enhancement of collection and acceleration of the wind. Besides, the blade tip vortices and the induced vortices weaken rapidly with their interaction, while the blade tip vortex without the brimmed diffuser remains even in the far downstream region

(Abe et al., 2009; Takahashi et al., 2012). Despite these researches about the rotor effect on the WLTs, it is still unknown how the rotor affects the frequency or intensity of the vortex shedding from the brimmed diffuser. For this reason, further research is required to investigate the rotor effect on VIV of the brimmed diffuser in the future.

We plan to add this explanation above to Sect. 1 in the revised manuscript, as written in **Revision** in **the 1st paragraph, Page 4**.

**5 Modal analysis**

**Comment**

"The value of the paper could be improved with experimental or operational modal analysis, as the authors appear to have access to a physical setup. This is not difficult to do and would provide a very useful validation of the numerical modal analysis."

**Response**

As the referee commented, our paper will be improved if experimental data about the natural frequency and the mode shape are added. However, unfortunately, we do not have data about the natural frequency, and the vibrational mode of the real wind lens model handled in the manuscript. It is because the wind lens turbine had already been demolished for another reason before this study began. However, the linear modal analysis for the 3D whole wind lens model with the accurate material properties will provide a reliable result without any major error, although a possible minor error in modal analysis would be often produced by the boundary condition, such as bolt joints. We plan to conduct an experiment to find the modal characteristics of a wind lens from the next study.

**6 Estimation of the critical wind speeds**

**Comment**

"I disagree with the statement that the presence of harmonics points to the presence of three separate vortex modes. Harmonics will appear in the FFT from the moment the variation the signal is not a perfectly sinusoidal function. A single mode is therefore perfectly possible."

**Response**

We used the term "vortex shedding mode" on the basis of the mode decomposition in the same ways as a natural (vibrational) mode. If mode deposition of the flow field in the frequency domain is carried out by Dynamic Mode Decomposition (DMD), separate vortex modes will clearly appear. According to the reports by Taira et al. (2020), Sakai et al. (2015), and Thomareis and Papadakis (2017), DMD for vortex shedding around a circular cylinder or an airfoil shows that each harmonic component of aerodynamic force corresponds to each vortex mode. As shown in Fig. 11 in our manuscript, it is visually evident that there are two pairs of vortices ($V_1^+$, $V_1^-$, $V_2^+$, $V_1^-$) generated over one cycle of the first vortex shedding frequency. These four vortices are related to the second vortex mode on the basis of the mode decomposition. In accordance with the reviewer's comment, we add the explanation of the vortex mode as below.

**Comment**

"There also appears to be little basis for the statement the first purported mode is primarily related to lift and the others to drag. Every vortex shedding mode will have a signature in both the lift and drag forces, even though obviously the period of drag variations is only half the period of the lift variations."

**Response**

The tendency of vortex shedding from the brimmed-diffuser shape is similar to that from an airfoil after stall at a large angle of attack rather than an ordinary bluff body. In the case of a circular cylinder or a square prism, the fluctuation frequency of drag force is twice as large as that of lift force because of the alternating vortices shedding from the upper and lower surfaces. On the other hand, in case of an airfoil after stall, the fluctuation frequency of lift and drag tend to be identical because the vortex shedding from the leading- and trailing edges occurs on only the upper (or suction) surface (Swalwell et al., 2003; Kurtulus, 2015). In the same manner, the brimmed-diffuser section generates a large lift force (mean Lift/Drag = 1.7) like an airfoil, and the vortices shed from only the inside (or suction) surface of the brimmed-diffuser section. Hence, it is not obvious that the fluctuation frequency of the drag is twice as large as that of the lift for airfoils and the brimmed-diffuser section. On the basis of the mode decomposition, harmonics of the lift and drag can be related to the respective vortex modes. From the peak intensities of frequency spectra of the lift and drag shown in Fig. 10b, the first vortex shedding mode is associated with the lift, and the second and third vortex shedding modes are mored associated with the drag than the lift.

**Revision**

Considering the reviewer's comment, we modify the explanation of the vortex mode in the revised manuscript as follows:

**Line 268–278, Page 12:** The sentence, "Both forces … the drag", is replaced as the follows.

The ratio of mean lift to drag is 1.7, which means that the aerodynamic characteristics of the brimmed-diffuser shape resemble those of an airfoil with circulation rather than a bluff body. The frequency spectrum of the lift and drag coefficients converted from their time histories by using Fast Fourier Transform (FFT) is shown in Fig. 10b. Figure 10b shows that lift and drag fluctuation includes three harmonic frequencies; the first Strouhal number ($St_1$) is 0.194, the second one ($St_2$) is 0.389, and the third one ($St_3$) is 0.584. Under the identical Reynolds number condition, the first Strouhal number of 0.194 for the wind lens is slightly smaller than 0.2016 for the cylinder (Norberg, 2003), larger than 0.147 for the square cylinder (Bai and Alam, 2018), and larger than 0.160 for the flat plate (Chen and Fang, 1996). According to Taira et al. (2020), Sakai et al. (2015), and Thomareis and Papadakis (2017), the dynamic mode decomposition (DMD) of a flow field in the frequency domain shows that each harmonic component for vortex shedding around a circular cylinder or an airfoil corresponds to each vortex mode. Therefore, the high harmonics included in the aerodynamic forces shown in Fig. 10b have independent vortex modes corresponding to the frequencies. Comparing to the peak intensity of each vortex shedding frequency, the first vortex mode is related to the lift dominantly, whereas the second and third vortex modes are related to the drag mainly.

**Line 299, Page 14:** The following sentences are added.

In summary, Fig. 11 shows that there are two pairs of vortices ($V_1^+$, $V_1^-$, $V_2^+$, $V_1^-$) generated over one cycle of the first vortex shedding frequency. These four vortices are related to the second vortex mode on the basis of the mode decomposition in the frequency domain.

**References**

Swalwell, K., Sheridan, J., and Melbourne, W.: Frequency Analysis of Surface Pressures on an Airfoil After Stall, 21st AIAA Applied Aerodynamics Conference, AIAA-2003-3416, doi: 10.2514/6.2003-3416, 2003.

Kurtulus, D. F.: On the Unsteady Behavior of the Flow around NACA 0012 Airfoil with Steady External Conditions at Re=1000, International Journal of Micro Air Vehicles, 7(3), doi: 10.1260/1756-8293.7.3.301, 2015.

**References to be added**

Taira, K., Hemati, M. S., Brunton, S. L., Sun, Y., Duraisamy, K., Bagheri, S., Dawson, S. T. M., and Yeh, C. A.,: Modal Analysis of Fluid Flows: Applications and Outlook, AIAA Journal, 58(3), doi: 10.2514/1.J058462, 2020

Sakai, M., Sunada, Y., Imamura, T., and Rinoie, K.: Experimental and Numerical Studies on Flow behind a Circular Cylinder Based on POD and DMD, Trans. Japan Soc. Aero. Space Sci., 58(2), pp. 100-107, doi: 10.2322/tjsass.58.100, 2015.

Thomareis, N. and Papadakis, G.: Effect of trailing edge shape on the separated flow characteristics around an

airfoil at low Reynolds number: A numerical study, Phys. Fluids, 29, 014101, doi: 10.1063/1.4973811, 2017.

Chen, J.M., and Fang, Y.C.: "Strouhal numbers of inclined flat plates," Journal of Wind Engineering and Industrial Aerodynamics, 61 (2), pp. 99-112, doi: 10.1016/0167-6105(96)00044-X, 1996.

**7 Lock-in**

**Comment**

"The lock-in phenomenon deserves a more thorough discussion. Its identification in fig. 16 is not convincing."

**Response**

In the current manuscript, we mentioned that we could not obtain the convergent results at $U^* = 5.1$, 5.3, and 5.5 because of the large deformation of the structure occurring in the transient response. Fortunately, we now obtained the limit-cycle oscillation results at $U^* = 5.1$ and 5.3 by gradually increasing the uniform wind speed, yet we could not obtain a convergent result at $U^* = 5.5$ because of much larger deformation in the transient response. To present the lock-in more clearly, we plan to replace Figs. 14, 15 and 16 with updated ones, as shown below. In the updated Fig. 16, we arrange the spectrum line of lift and drag at the respective velocities to accurately express the discretized data instead of the surface contour of frequency spectra in the whole calculation domain. Even if the figures are updated in the revised version, the explanations related to Figs. 14 and 15 are the same as those in the current manuscript because the tendency of the graphs is the same. Thanks to the updated figures, we will be able to add the discussion about the lock-in phenomena in more detail in the revised version.

**Revision**

[Figure]

**Figure 14. Compliance of each mode with respect to wind speed in the aeroelastic analysis**

[Figure]

**Figure 15. The displacement amplitude of the brim tip in the x-direction and the throat in the r-direction**

[Figure]

(a) Frequency spectra of lift

(b) Frequency spectra of drag

**Figure 1. Frequency spectra of lift and drag at each reduced velocity in the aeroelastic analysis**

**Lines 401–409, Page 19:** The sentence, "The intensities … amplitude vibration", is replaced to the following

The intensities of the lift and drag spectra with respect to the reduced velocity are expressed in Fig. 16. As aforementioned, the result at $U^* = 5.5$, where the computation was aborted due to the excessive deformation in the transient response, is blank. The peak points of the spectra are aligned with the respective straight vortex-shedding frequency lines estimated from the Strouhal numbers in the case where the rigid wind lens section is at rest. However, in the range of $U^* = 4.9$–$5.5$, the peak points are located along the respective horizontal natural frequency lines, which indicates that a lock-in phenomenon occurs. At $U^* = 4.9$, the second vortex shedding frequency is captured in the third natural frequency. Nevertheless, the third mode vibration is not excited so much as shown in Fig. 14. At $U^* = 5.1$ and $5.3$, when the first mode vibration is considerably excited as shown in Fig. 14, the first vortex shedding frequency is locked in the first natural frequency. At $U^* = 5.5$ (blank), the first vortex shedding frequency is likely to match the second natural frequency, which can be inferred from the additional peaks of the lift and drag spectra that appear on the horizontal line of $f^* = f^*_{n2}$ at $U^* \geq 5.7$. On the other hand, there is no noticeable lock-in at $U^* = 2.5$ and $2.8$ at the intersections between the second vortex shedding frequency and the

first and second natural frequencies, although the substantial vibration occurs as shown in Fig. 15. According to Kumar et al. (2016), the width in frequency where lock-in occurs is dependent on $A/H$ for a circular cylinder. From Figs. 15 and 16, the difference between lock-in at $U^* = 4.9$ and non-lock-in at $U^* = 2.8$ is attributed to $A/H$ = 0.04 and 0.01, respectively.

**References to be added**

Kumar, S., Navrose, and Mittal, S.: Lock-in in forced vibration of a circular cylinder, Phys. Fluids, 28, 113605, doi: 10.1063/1.4967729, 2016.

**9 Minor comments**

**Comment**

"The authors use a compressible solver in the CFD. Why do they expect compressibility to be important?"

**Response**

According to the literature (Shima, 2015), numerical errors due to the low Mach number increase with decreasing a Mach number less than 0.1 without any preconditioner. In the CFD program, the Mach number was set to 0.1 imaginarily. We confirmed that the calculation results are almost the same with less than 1% error in the Strouhal numbers at the Mach number set as 0.1, 0.2, and 0.3. The CFD program has been validated in several unsteady aerodynamic studies at low Reynolds number regime. In these studies (Isogai et al, 2004; Yamamoto and Isogai, 2005; Nagai et al, 2009; Nagai and Isogai, 2011; Nagai et al, 2012), the numerical results are in good agreement with the experimental results. The explanation for the compressible solver is added to the revised version.

**Revision**

**Line 190, Page 9:** we plan to add the sentences below at the end of Sect. 3.2.

Additionally, the Mach number in this CFD program was set to 0.1 imaginarily. We confirmed that the calculation results are almost the same with less than 1% error in the Strouhal numbers at Mach numbers of 0.1, 0.2, and 0.3. Also, we compared a numerical result calculated by 2D Reynolds-Averaged Navier-Stokes (RANS) turbulence model simulation with the SST $k$-$\omega$ turbulence model, utilizing the commercial software, ANSYS CFX. The details are discussed in Appendix B. The vortex shedding frequencies and the vortex intensities in both results are in good agreement.

**References**

Shima, E.: Simple Compressible CFD Solvers' Story, Nagare : Journal of Japan Society of Fluid Mechanics, 34(2), 67-79, 2015 (in Japanese).

Isogai, K., Fujishiro, S., Saitoh, T., Yamamoto, M., Yamasaki, M., and Matsubara, M.: Unsteady Three-Dimensional Viscous Flow Simulation of a Dragonfly Hovering, AIAA J., 42(10), 2053-2059, doi: 10.2514/1.6274, 2004.

Yamamoto, M. and Isogai, K.: Direct Measurement of Unsteady Fluid Dynamic Forces for a Hovering Dragonfly, AIAA J., 43(12), 2475-2480, doi: 10.2514/1.15899, 2005.

Nagai, H., Isogai, K., Fujimoto, T. and Hayase, T.: Experimental and Numerical Study of Forward Flight Aerodynamics of Insect Flapping Wing, AIAA J., 47(3), 730–742, doi:10.2514/1.39462, 2009.

Nagai, H. and Isogai, K.: Effects of Flapping Wing Kinematics on Hovering and Forward Flight Aerodynamics, AIAA J., 49(8), 1750-1762, doi: 10.2514/1.J050968, 2011.

Nagai, H., Isogai, K., Murozono, M. and Fujishiro, T.: INVESTIGATION ON STRUCTURAL AND

AERODYNAMIC CHARACTERISTICS OF RESONANT TYPE ELASTIC FLAPPING WING, in 28th Congress of the International Council of the Aeronautical Sciences, p. ICAS 2012-9.5.3, Brisbane, Australia., 2012. http://www.icas.org/ICAS_ARCHIVE/ICAS2012/ABSTRACTS/875.HTM

**Comment**

"Two letter symbols such as St and Re should not be italic."

**Response**

We consider the Strouhal numbers and the Reynolds numbers as variables. For this reason, $St$ and $Re$ are expressed as the italic font in the manuscript. Below are the publications that we referred to.

Bearman, P. W., Gartshore, I. S., Maull, D. J. and Parkinson, G. V.: Experiments on flow-induced vibration of a square-section cylinder, J. Fluids Struct., 1(1), 19–34, doi:10.1016/S0889-9746(87)90158-7, 1987.

Bai, H. and Alam, M. M.: Dependence of square cylinder wake on Reynolds number, Phys. Fluids, 30(1), doi:10.1063/1.4996945, 2018.

https://wes.copernicus.org/preprints/wes-2020-40/

https://wes.copernicus.org/preprints/wes-2019-83/

There are two WES manuscripts above, and we decided to match the font of $St$ and $Re$ with other manuscripts in WES. In our manuscript, we found that some remained the normal font in Figs. 10, 13, and A 1, and we plan to italicize them.

**Other revisions**

**Line 372, Page 19:** The sentence, "As shown in Fig. 14, the third and fourth natural modes are excited by the weak second vortex mode only, while the responses of the first and second natural modes are noticeable because the radial and rotational modes are more sensitive to the three vortex modes.", will be replaced for the clear and concise explanation as follows.

As shown in Fig. 14, the camber-bending and horizontal modes are hard to be excited due to the higher modal stiffness, while the responses of the radial and rotational modes are noticeable because of the lower modal stiffness and the sensitivity of the vortex modes.

**Line 428-433, Page 21:** The sentences, "The two critical modes are excited by the first vortex mode at the wind speed of 24 to 27 m/s ($U^*$= 4.87 to 5.72), which results in considerably large oscillation. Also, the second vortex mode excites the radial and rotational modes at 12 and 13 m/s ($U^*$= 2.54 to 2.76), respectively, which causes noticeable oscillation with a relatively small amplitude. These wind speeds of 12 and 13 m/s are close to the average wind speed when the actual VIV of the brimmed diffuser was observed. Although the 2D aeroelastic analysis neglects some complex phenomena that occur in the actual wind lens, it provides a reasonable estimation of the critical wind speeds for VIV in the wind lens structure.", will be replaced because the lock-in is shown in the renewed Fig. 16.

The two critical vibrational modes are excited by the first vortex mode at the wind speed of $U^*$= 5.30 to 5.72, at which the first vortex frequency is locked in the first or second natural frequency. This lock-in results in large oscillation. Also, the second vortex mode excites the radial and rotational modes at $U^*$= 2.54 and 2.75, respectively, which causes noticeable oscillation with a relatively small amplitude. In this reduced-velocity range, no lock-in occurs. Although the 2D aeroelastic analysis neglects some complex phenomena that occur in the actual wind lens, it provides a reasonable estimation of the critical wind speeds for VIV in the wind lens structure.

---

## Author Comment (AC2) · 15 Jul 2020

**Authors' Responses to the comments of Referee #2 on**

"Fundamental effect of vibrational mode on vortex-induced vibration in a brimmed diffuser for a wind turbine" by Taeyoung Kim et al. (wes-2020-29)

We are grateful for your valuable comments that help us improve the quality of this manuscript. We replied to your comments as below.

**Comment 1**

"Authors decided to focus on 2D simulations, which is reasonable due to the high computational cost of aero-structural simulations, however more specific justification on the technical relevance of this choice is required."

**Response**

This study intends to explain and clarify the fundamental mechanism of the vortex-induced vibration (VIV) in the wind lens structure rather than to perform the realistic simulation of the flow-induced vibration (FIV) of the actual wind lens. In order to obtain a fundamental understanding of the VIV in wind lens and simplify the discussion, we minimized the turbulent effect, 3D wake structure, and the rotor effect in the analysis. On the basis of the reviewer's comment, We plan to add supplementary explanations to the first paragraph on Page 4 in the introduction of the revised manuscript to emphasize the novelty of this paper, including the reason why we reduced our numerical model to the two-dimension and the low Reynolds number, which is written in **Response** to **Comment 5**.

**Comment 2**

"Authors make use of a compressible-flow model. Since the Mach number should be well below 0.3, this choice does not seem justified and might even lead to numerical issues if proper preconditioners are not used. Please clarify in the text."

**Response**

According to the literature (Shima, 2015), numerical errors due to the low Mach number increase with decreasing a Mach number less than 0.1 without any preconditioner. In our CFD program, the Mach number was set as 0.1 imaginarily. We confirmed that the calculation results are almost the same with less than 1% error in the Strouhal numbers at the Mach number set as 0.1, 0.2, and 0.3. Our CFD program has been validated in several unsteady aerodynamic studies at low Reynolds number regime. In these studies (Isogai et al, 2004; Yamamoto and Isogai, 2005; Nagai et al, 2009; Nagai and Isogai, 2011; Nagai et al, 2012), the numerical results are in good agreement with the experimental results. In addition, as shown below in **Response** to **Comment 6**, the numerical result is compared to that calculated by using ANSYS CFX with the SST $k$-$\omega$ turbulence model at $Re$ = 288. The vortex shedding frequencies and the vortex intensity in both results are in good agreement.

**Revision**

**Line 190, Page 9:** we plan to add the sentences below at the end of Sect. 3.2.

Additionally, the Mach number in this CFD program was set to 0.1 imaginarily. We confirmed that the calculation results are almost the same with less than 1% error in the Strouhal numbers at Mach numbers of 0.1, 0.2, and 0.3. Also, we compared a numerical result calculated by 2D Reynolds-Averaged Navier-Stokes (RANS) turbulence model simulation with the SST $k$-$\omega$ turbulence model, utilizing the commercial software, ANSYS CFX. The details are discussed in Appendix B. The vortex shedding frequencies and the vortex intensities in both results are

in good agreement.

**References**

Shima, E.: Simple Compressible CFD Solvers' Story, Nagare : Journal of Japan Society of Fluid Mechanics, 34(2), 67-79, 2015 (in Japanese).

Isogai, K., Fujishiro, S., Saitoh, T., Yamamoto, M., Yamasaki, M., and Matsubara, M.: Unsteady Three-Dimensional Viscous Flow Simulation of a Dragonfly Hovering, AIAA J., 42(10), 2053-2059, doi: 10.2514/1.6274, 2004.

Yamamoto, M. and Isogai, K.: Direct Measurement of Unsteady Fluid Dynamic Forces for a Hovering Dragonfly, AIAA J., 43(12), 2475-2480, doi: 10.2514/1.15899, 2005.

Nagai, H., Isogai, K., Fujimoto, T. and Hayase, T.: Experimental and Numerical Study of Forward Flight Aerodynamics of Insect Flapping Wing, AIAA J., 47(3), 730–742, doi:10.2514/1.39462, 2009.

Nagai, H. and Isogai, K.: Effects of Flapping Wing Kinematics on Hovering and Forward Flight Aerodynamics, AIAA J., 49(8), 1750-1762, doi: 10.2514/1.J050968, 2011.

Nagai, H., Isogai, K., Murozono, M. and Fujishiro, T.: INVESTIGATION ON STRUCTURAL AND AERODYNAMIC CHARACTERISTICS OF RESONANT TYPE ELASTIC FLAPPING WING, in 28th Congress of the International Council of the Aeronautical Sciences, p. ICAS 2012-9.5.3, Brisbane, Australia., 2012. http://www.icas.org/ICAS_ARCHIVE/ICAS2012/ABSTRACTS/875.HTM

**Comment 3**

"The authors use a deforming mesh strategy to account for the deformation of the structure during motion; some information on the numerical technique used for deforming the mesh should be reported (at least with a citation). Did the Authors experience any issue with cell quality due to mesh deformation during the simulations?"

**Response 1**

The sentence below will be added to the revised version to explain the moving mesh method in **Line 167, Page 7**.

"The moving mesh method based on the geometric conservative law was used (Thomas and Lombard, 1979)."

**Reference to be added**

Thomas, P. D., and Lombard, C. K.: Geometric Conservation Law and Its Application to Flow Computations on Moving Grids, AIAA Journal, 17 (10), 1030–1037, doi: 10.2514/3.61273, 1979.

**Response 2**

In the current manuscript, we mentioned that we could not obtain the convergent results at $U^* = 5.1$, 5.3, and 5.5 because of the large deformation of the structure occurring in the transient response. Fortunately, we now obtained the limit-cycle oscillation results at $U^* = 5.1$ and 5.3 by gradually increasing the uniform wind speed, yet we could not obtain a convergent result at $U^* = 5.5$ because of much larger deformation in the transient response. To present the lock-in more clearly, we plan to replace Figs. 14, 15 and 16 with updated ones, as shown below. In the updated Fig. 16, we arrange the spectrum line of lift and drag at the respective velocities in order to accurately express the discretized data instead of the surface contour of frequency spectra in the whole calculation domain. Even if the figures are updated in the revised version, the explanations related to Figs. 14 and 15 are the same as those in the current manuscript because the tendency of the graphs is the same. Thanks to the updated figures, we add the discussion about the lock-in phenomena in more detail in the revised version.

[Figure]

**Figure 14. Compliance of each mode with respect to wind speed in the aeroelastic analysis**

[Figure]

**Figure 15. The displacement amplitude of the brim tip in the x-direction and the throat in the r-direction**

[Figure]

**Figure 16. Frequency spectra of lift and drag at each reduced velocity in the aeroelastic analysis**

**Lines 401–409, Page 19:** The sentence, "The intensities … amplitude vibration", is replaced to the following

The intensities of the lift and drag spectra with respect to the reduced velocity are expressed in Fig. 16. As aforementioned, the result at $U^* = 5.5$, where the computation was aborted due to the excessive deformation in the transient response, is blank. The peak points of the spectra are aligned with the respective straight vortex-shedding frequency lines estimated from the Strouhal numbers in the case where the rigid wind lens section is at rest. However, in the range of $U^* = 4.9$–5.5, the peak points are located along the respective horizontal natural frequency lines, which indicates that a lock-in phenomenon occurs.   At $U^* = 4.9$, the second vortex shedding frequency is captured in the third natural frequency. Nevertheless, the third mode vibration is not excited so much as shown in Fig. 14. At $U^* = 5.1$ and 5.3, when the first mode vibration is considerably excited as shown in Fig. 14, the first vortex shedding frequency is locked in the first natural frequency. At $U^* = 5.5$ (blank), the first vortex shedding frequency is likely to match the second natural frequency, which can be inferred from the additional peaks of the lift and drag spectra that appear on the horizontal line of $f^* = f^*_{n2}$ at $U^* \geq 5.7$. On the other hand, there is no noticeable lock-in at $U^* = 2.5$ and 2.8 at the intersections between the second vortex shedding frequency and the first and second natural frequencies, although the substantial vibration occurs as shown in Fig. 15. According to Kumar et al. (2016), the width in frequency where lock-in occurs is dependent on $A/H$ for a circular cylinder. From Figs. 15 and 16, the difference between lock-in at $U^* = 4.9$ and non-lock-in at $U^* = 2.8$ is attributed to $A/H = 0.04$ and 0.01, respectively.

**References to be added**

Kumar, S., Navrose, and Mittal, S.: Lock-in in forced vibration of a circular cylinder, Phys. Fluids, 28, 113605, doi: 10.1063/1.4967729, 2016.

**Comment 4**

"No model of the turbine within the wind lens is considered, at least a comment on this aspect should be reported."

**Response**

The consideration of the rotor is important to a realistic full simulation of VIV for the actual wind lens. However, to minimize the aerodynamic complexity and elicit the fundamental mechanism of VIV for the brimmed diffuser, we removed the rotor from the numerical model. We plan to add the survey of the previous studies about the rotor

effect on the wind lens turbines and the explanation for removing the rotor from our numerical model in the Introduction of the revised manuscript.

The corresponding revision is written in the third paragraph of **Revision** of **Comment 5** below in addition to the other revision of the Introduction.

**Comment 5**

"Authors decided to reduce by three orders of magnitude the Reynolds number of the problem, altering the flow regime from turbulent (the actual one) to laminar. Authors state that similarity in aero-elastic response is confirmed despite the alteration of the Reynolds number, but they do not explain how and why. As well known, turbulence is a non-linear phenomenon which does not only feature unsteadiness and instability, but which alters the gross properties of the flow and, hence, the values of lift and drag coefficients, as well as the vortex shedding frequency. Why do the Authors believe that such an arbitrary change in flow regime does not lead to unrealistic results?"

**Response**

As mentioned in **Response** to **Comment 1**, the objective of this study is to reveal the fundamental mechanism of VIV in the wind lens structure with a 2D simplified analytical model. Unintendedly, a lack of explanation and some of words might have led readers to misunderstanding the object of this study. We plan to add supplementary explanations to the Introduction of the revised manuscript to emphasize the novelty of this paper, including the reason why we reduced our numerical model to the two-dimension and the low Reynolds number, as follows.

**Revision**

**The 1st paragraph on Page 4:** The sentence, "Motivated by … numerical simulation", is replaced to the following

Despite many previous studies of VIV for bluff bodies, there is little fundamental understanding of VIV of the brimmed diffuser, whose aerodynamic characteristics contain those of both bluff bodies and airfoils. The brimmed-diffuser section always generates a larger mean lift force with circulation than a mean drag force. On the contrary, the general bluff bodies, such as a circular or rectangular cylinder, generate lift and drag fluctuation, in which the lift force averages zero over time. From this point of view, the aerodynamics of the brimmed-diffuser section resembles that of an airfoil with a large flap angle at a high angle of attack after stall, rather than that of the bluff bodies. Recently, some studies of VIV for a simple airfoil without a flap (Skrzypiński et al., 2014; Benner et al., 2019) have been conducted with the interest of VIV for a wind turbine blade in a standstill condition. Nonetheless, the VIV of an airfoil with a large flap after stall, which is similar to the brimmed-diffuser shape, has not received attention because it does not happen in general operation for aircraft and wind turbines.

Motivated by the VIV observed in the actual wind lens, we numerically investigated the fundamental mechanism of VIV for the brimmed-diffuser shape. Considering the average wind speed on April 3, 2012 and the size of the wind lens, the wind lens turbine was in operation at a Reynolds number of approximately 300,000. The Reynolds number near 300,000 is in the critical Reynolds number regime, where the laminar boundary layer of a bluff body undergoes a turbulent transition, and the wake is disorganized and has 3D vortex structures (Williamson, 1996; Anderson, 2007). Furthermore, the vortex shedding frequency is sensitive to the surface condition of the body and the turbulence intensity of the flow (Zdravkovich, 1990). Because of the complexity of the turbulence and high computational cost, the precise numerical simulation of VIV in the large Reynolds number regime is still challenging even for a circular cylinder (Sarpkaya, 2004; Nguyen and Nguyen, 2016).

Another factor to complicate the flow around the brimmed diffuser is a rotor effect. According to Hasegawa et al. (2007), the rotor inside the wind lens causes a three-dimensional flow around the wind lens, whereas the flow around the wind lens without a rotor tends to be two-dimensional. In addition, the vortex generated by the rotor blade tip induces another vortex in the boundary layer of the inner surface of the wind lens, and the induced vortex suppresses flow separation from the inner surface of the wind lens, which leads to enhancement of collection and

acceleration of the wind (Takahashi et al., 2012). Then, the blade tip vortices and the induced vortices rapidly weaken with their interaction, while the blade tip vortex without the brimmed diffuser remains even in the far downstream region (Abe et al., 2009; Takahashi et al., 2012). Despite these studies about the rotor effect on the WLTs, it has been unremarked how the rotor affects the frequency or intensity of the vortex shedding from the brimmed diffuser.

To initiate the fundamental investigation of VIV for the brimmed-diffuser shape, it is reasonable to simplify the analytical model by minimizing complexities, such as the turbulent effect, 3D wake structure, and the rotor effect. Therefore, the purpose of this paper is not to simulate the observed VIV for the actual wind lens, but rather to clarify the fundamental mechanism of VIV for the brimmed-diffuser shape. To elicit the fundamental mechanism of VIV, the 2D aerodynamic model is employed at a low Reynolds number of 288 with the rotor excluded, where the vortex shedding structures can be treated as 2D in the wake. The vibrational modal characteristics for the whole 3D wind lens are calculated by using the finite element method (FEM). Coupled with the equations of motion in the modal space, 2D unsteady aeroelastic simulation based on 2D Navier-Stokes equations is performed.

**References to be added**

Benner, B. M., Carlson, D. W., Seyed-Aghazadeh, B., and Modarres-Sadeghi, Y.: Vortex-Induced Vibration of Symmetric airfoils used in Vertical-Axis Wind Turbines, Journal of Fluids and Strucutures, 91, 102577, doi: 10.1016/j.jfluidstructs.2019.01.018, 2019.

Hasegawa, M., Ohya, Y. and Kume, H.: Numerical Studies of Flows Around a Wind Turbine Equipped with Flanged-Diffuser Shroud by Using an Actuator-Disc Model, Trans. Japan Soc. Mech. Eng. Ser. B, 73(733), 1860–1867, doi:10.1299/kikaib.73.1860, 2007. (in Japanese)

Skrzypiński, W. R., Gaunaa, M., Sørensen, N., Zahle, F., and Heinz, J.: Self-induced vibration of a DU96-W-180 airfoil in stall, Wind Energy, 17, pp. 641–655, doi: 10.1002/we.1596, 2014.

Takahashi, S., Hata, Y., Ohya, Y., Karasudani, T. and Uchida, T.: Behavior of the blade tip vortices of a wind turbine equipped with a brimmed-diffuser shroud, Energies, 5(12), 5229–5242, doi:10.3390/en5125229, 2012.

**Comment 6**

"with present-day CFD technology it is possible to simulate highly turbulent flows with U-RANS, employing suitable turbulence models (such Spalart-Allmaras or k-w SST models)"

**Response**

An additional 2D CFD for the rigid case at $Re$ = 288 was conducted by using ANSYS CFX with a turbulence model of SST $k$-$\omega$ to be compared with the result of our 2D CFD. The vortex shedding frequencies and the vortex intensity in both results are in good agreement.

**Revision**

**Appendix B:** We plan to add this result as Appendix B as below.

[Figure]

[Figure]

**Figure B 1. Comparison with the frequency spectrum of the lift and drag coefficients calculated by ANSYS CFX with the SST *k-ω* turbulence model**

Appendix B demonstrates a result of the frequency spectrum from the time history of the lift and drag coefficients calculated at $Re = 288$ by 2D RANS turbulence model simulation, using ANSYS CFX 19.2, to verify that the Strouhal numbers calculated the in-house CFD program are valid. In this case, the SST $k–\omega$ turbulence model was applied. The shape and boundary of the computation domain was the same as those in the in-house CFD shown in Fig. 4a, and its domain size was $27.2L \times 13.6L$. The wind lens section is placed where its throat is $0.5D_{thr}$ above the bottom of the domain and $4.5L$ horizontally away from the left side of the domain. The total number of nodes was 490,142 and the number of elements was 244,175. The time step was 0.001 and the value of $y^+$ was approximately 1. The calculation was conducted until the 70th cycle of the first vortex frequency. Figure B1 shows the frequency spectra of lift and drag calculated by the two different CFD solvers. The two results show a good agreement in the Strouhal numbers although there is a slight difference in the intensity of the respective peaks.

---

## Editor Comment (EC2) · Alessandro Bianchini (Editor) · 16 Jul 2020

Dear authors, I have gone through your replies to Reviewers' comments. Please upload the revised version of the paper according to the requested modifications. Since relevant doubts were expressed by the Reviewers about the soudness of your methodology, the new version of the study will be revised again by me and the Reviewers. Thus, please consider that a resubmission does not ensure a final publication in WES. Best regards,
* * *